# Temperatures from Energy Balance Models: the effective heat capacity matters

Gerrit Lohmann[1,2]

[1]Alfred Wegener Institute, Helmholtz Centre for Polar and Marine Research, Bremerhaven, Germany
[2] University of Bremen, Bremen, Germany

**Correspondence:** Gerrit Lohmann (Gerrit.Lohmann@awi.de)

**Abstract.** Energy balance models (EBM) are highly simplified models of the climate system, providing admissible conceptual tools for understanding climate changes. The global temperature is calculated by the radiation budget through the incoming energy from the Sun and the outgoing energy from the Earth. The argument that the temperature can be calculated by this simple radiation budget is revisited. The underlying assumption for a realistic temperature distribution is explored: One has to assume a moderate diurnal cycle due to the large heat capacity and the fast rotation of the Earth. Interestingly, the global mean in the revised EBM is very close to the originally proposed value. The main point is, that the effective heat capacity and its temporal variation over the daily/seasonal cycle needs to be taken into account when estimating surface temperature from the energy budget. Furthermore, the time dependent-EBM predicts a flat meridional temperature gradient for large heat capacities, reducing the seasonal cycle, reducing the outgoing radiation and increasing global temperature. Motivated by this finding, a sensitivity experiment with a complex model is performed where the vertical diffusion in the ocean has been increased. The resulting temperature gradient, reduced seasonal cycle, and global warming is also found in climate reconstructions, providing a possible mechanism for past climate changes prior to 3 million years ago.

**Keywords.** Energy balance model, Earth system modeling, Temperature gradient, Climate change, Climate sensitivity, Climate reconstructions

## 1 Introduction

Energy balance models (EBMs) are among the simplest climate models. They were introduced almost simultaneously by Budyko (1969) and Sellers (1969). Because of their simplicity, these models are easy to understand and facilitate both analytical and numerical studies of climate sensitivity (Peixoto and Oort, 1992; Hartmann, 1994; Saltzmann, 2001; Ruddiman, 2001; Pierrehumbert, 2010). A key feature of these models is that they eliminate the climate's dependence on the wind field, ocean currents, the Earth rotation, and thus have only one dependent variable: the Earth's near-surface air temperature T.

With the development of computer capacities, simpler models have not disappeared; on the contrary, a stronger emphasis has been given to the concept of a hierarchy of models' as the only way to provide a linkage between theoretical understanding and the complexity of realistic models (von Storch et al. 1999; Claussen et al. 2002). In contrast, many important scientific

debates in recent years have had their origin in the use of conceptually simple models (Le Treut et al., 2007; Stocker, 2011), also as a way to analyze data (Köhler et al., 2010) or complex models (Knorr et al., 2011).

Pioneering work has been done by North (North, 1975a, b; 1981; 1983) and these models were applied subsequently (e.g., Ghil, 1976; Su and Hsieh, 1976; Fraedrich, 1979; Ghil and Childress, 1987; Short et al., 1991; Stocker et al., 1992; North and Kim, 2017). Later the EMBs were equipped by the hydrological cycle (Chen et al., 1995; Lohmann et al., 1996; Fanning and Weaver, 1996; Lohmann and Gerdes, 1998) to study the feedbacks in the atmosphere-ocean-sea ice system. One of the most useful examples of a simple, but powerful, model is the one-/zero-dimensional energy balance model. As a starting point, a zero-dimensional model of the radiative equilibrium of the Earth is introduced (Fig. 1)

$$(1 - \alpha)S\pi R^2 = 4\pi R^2 \epsilon \sigma T^4 \tag{1}$$

where the left hand side represents the incoming energy from the Sun (size of the disk= shadow area $\pi R^2$) while the right hand side represents the outgoing energy from the Earth (Fig. 1). T is calculated from the Stefan-Boltzmann law assuming a constant radiative temperature, S is the solar constant - the incoming solar radiation per unit area– about $1367\,\mathrm{W m^{-2}}$, $\alpha$ is the Earth's average planetary albedo, measured to be 0.3. R is Earth's radius $= 6.371 \cdot 10^6$ m, $\sigma$ is the Stefan-Boltzmann constant $= 5.67 \cdot 10^{-8} \mathrm{JK^{-4}m^{-2}s^{-1}}$, and $\epsilon$ is the effective emissivity of Earth (about 0.612) (e.g., Archer 2010). The geometrical constant $\pi R^2$ can be factored out, giving

$$(1 - \alpha)S = 4\epsilon\sigma T^4 \tag{2}$$

Solving for the temperature,

$$T = \sqrt[4]{\frac{(1 - \alpha)S}{4\epsilon\sigma}} \tag{3}$$

Since the use of the effective emissivity $\epsilon$ in (1) already accounts for the greenhouse effect we gain an average Earth temperature of 288 K (15°C), very close to the global temperature observations/reconstructions (Hansen et al., 2011) at 14°C for 1951-1980. The implicit assumption in (2,3) is that we have a single temperature on the Earth, although we know that the equator-to-pole surface temperature gradient is in the order of 50 K, and that the incoming solar radiation at the equator is about twice that at the poles (Peixoto and Oort, 1992).

Furthermore, (3) does not contain parameters like the heat capacity of the planet. We will explore that this is essential for the temperature of the Earth's climate system. We will evaluate the effect of the effective heat capacity in the climate system. Schwartz (2007) stressed out that the effective heat capacity is not an intrinsic property of the climate system but is reflective of the rate of penetration of heat energy into the ocean in response to the particular pattern of forcing and background state.

Wang et al. (2019) showed a pronounced low equator-to-pole gradient in the annual mean sea surface temperatures is found in a numerical experiment conducted with a coupled model consisting of an atmospheric general circulation model coupled to a slab ocean model in which the mixed-layer thickness is reduced. In the present paper, it is shown that the heat capacity is linked to the question of a low equator-to-pole gradients during the Paleogene/Neogene climate (Markwick, 1994; Wolfe, 1994; Sloan and Rea, 1996; Huber et al., 2000; Shellito et al., 2003; Tripati et al., 2003; Mosbrugger et al., 2005). Those published temperature patterns resemble the high latitude warming (with moderate low latitude warming) and reduced seasonality.

## 2 Results

### 2.1 The spatial distribution of the energy balance

Let us have a closer look onto (1) and consider *local* radiative equilibrium of the Earth at each point. Fig. 2 shows the latittude-longitude dependence of the incoming short wave radiation. The global mean temperatures are not affected by the tilt (Berger and Loutre 1991; 1997; Laepple and Lohmann 2009). We assume an idealized geometry of the Earth, no obliquity and no precession, which makes an analytical calculation possible. Furthermore $\epsilon$ and $\alpha$ are assumed to be constants.

The incoming radiation goes with the cosine of latitude $\varphi$ and longitude $\Theta$, and there is only sunshine during the day. Fig. 2a shows the latitudinal dependence. As we assume no tilt (this assumption is later relaxed), the latitudinal dependence is a function of latitude only: $\cos\varphi$. On the right-hand side, the function is shown. Fig. 2b shows the latitudinal dependence is a function of longitude: $\cos\Theta$ for the sun-shining side of the Earth, and for the dark side of the Earth it is zero. For simplicity, we can define the angle $\Theta$ anti-clockwise on the for the sun-shining side between $-\pi/2$ and $\pi/2$. We define the maximal insolation always at $\Theta = 0$ which is moving in time. In the panel, the Earth's rotation is schematically sketched as the red arrow, and we see the time-dependence in the right-hand side. It is noted that the geographical longitude can be calculated by $mod(\Theta - 2\pi \cdot t/24, 2\pi)$ where t is measured in hours and mod is the modulo operation. Summarizing our geometrical considerations, we can now write the local energy balance as

$$
\begin{aligned}
\epsilon\sigma T^4 &= (1-\alpha)S \cdot \cos\varphi \cdot \cos\Theta \qquad \text{for } -\pi/2 < \Theta < \pi/2. \\
&= 0 \qquad\qquad\qquad\qquad\qquad \text{elsewhere}
\end{aligned}
\tag{4}
$$

Temperatures based on the local energy balance without a heat capacity would vary between $T_{min} = 0$ K and $T_{max} = \sqrt[4]{\frac{(1-\alpha)S}{\epsilon\sigma}} = \sqrt{2} \cdot \sqrt[4]{\frac{(1-\alpha)S}{4\epsilon\sigma}} = \sqrt{2} \cdot 288 K = 407$ K. Integration of (4) over the Earth surface is

$$
\int_{-\pi/2}^{\pi/2} \left( \int_0^{2\pi} \epsilon\sigma T^4 R\cos\varphi d\Theta \right) R d\varphi = (1-\alpha)S \int_{-\pi/2}^{\pi/2} R\cos^2\varphi d\varphi \cdot \int_{-\pi/2}^{\pi/2} R\cos\Theta d\Theta
$$

$$
\epsilon\sigma R^2 \frac{4\pi}{4\pi} \int_{-\pi/2}^{\pi/2} \left( \int_0^{2\pi} T^4 \cos\varphi d\Theta \right) d\varphi = (1-\alpha)SR^2 \underbrace{\int_{-\pi/2}^{\pi/2} \cos^2\varphi \, d\varphi}_{\frac{\pi}{2}} \cdot \underbrace{\int_{-\pi/2}^{\pi/2} \cos\Theta \, d\Theta}_{2}
$$

$$
\epsilon\sigma 4\pi \overline{T^4} = (1-\alpha)S\,\pi
\tag{5}
$$

giving a similar formula as (3) with the definition for the average

$$
\overline{T^4} = \frac{1}{4\pi} \int_{-\pi/2}^{\pi/2} d\varphi \int_0^{2\pi} \cos\varphi \quad d\Theta \quad T^4 \quad .
$$

What we really want is the mean of the temperature $\overline{T}$. Therefore, we take the fourth root of (4):

$$
T = \sqrt[4]{\frac{(1-\alpha)S \cos\varphi \cos\Theta}{\epsilon\sigma}} \qquad \text{for } -\pi/2 < \Theta < \pi/2 \qquad \text{and zero elsewhere.}
\tag{6}
$$

If we calculate the zonal mean of (6) by integration at the latitudinal cycles we have

$$T(\varphi) = \frac{\sqrt{2}}{2\pi} \underbrace{\int_{-\pi/2}^{\pi/2} (\cos\Theta)^{1/4} d\Theta}_{\sqrt{\pi}\Gamma(5/8)/\Gamma(9/8)} \sqrt[4]{\frac{(1-\alpha)S}{4\epsilon\sigma}} (\cos\varphi)^{1/4} = \underbrace{\frac{1}{\sqrt{2\pi}} \frac{\Gamma(5/8)}{\Gamma(9/8)}}_{\approx 0.608} \cdot \sqrt[4]{\frac{(1-\alpha)S}{4\epsilon\sigma}} (\cos\varphi)^{1/4} \tag{7}$$

as a function on latitude (Fig. 3). $\Gamma$ is Euler's Gamma function with $\Gamma(x+1) = x\Gamma(x)$. When we integrate this over the latitudes,

$$\overline{T} = \frac{1}{2} \int_{-\pi/2}^{\pi/2} T(\varphi)\cos\varphi \, d\varphi = \frac{1}{2} \frac{\Gamma(5/8)}{\sqrt{2\pi}\Gamma(9/8)} \cdot \sqrt[4]{\frac{(1-\alpha)S}{4\epsilon\sigma}} \underbrace{\int_{-\pi/2}^{\pi/2} (\cos\varphi)^{5/4} d\varphi}_{\sqrt{\pi}\Gamma(9/8)/\Gamma(13/8)} = \underbrace{\frac{1}{2}\frac{1}{\sqrt{2}}\frac{\Gamma(5/8)}{\Gamma(13/8)}}_{\frac{\sqrt{2}}{4}\frac{8}{5}=0.4\sqrt{2}} \cdot \sqrt[4]{\frac{(1-\alpha)S}{4\epsilon\sigma}} \quad . \tag{8}$$

5    Therefore, the mean temperature is a factor $F = 0.4\sqrt{2} \approx 0.566$ lower than 288 K as stated at (3) and would be $\overline{T} = 163$ K. The standard EBM in Fig. 1 has imprinted into our thoughts and lectures. We should therefore be careful and pinpoint the reasons for the failure. What happens here is that the heat capacity of the Earth is neglected and there is a strong non-linearity of the outgoing radiation. The local radiative equilibrium assumption cannot be used at these conditions.

## 2.2 The heat capacity and fast rotating body

10    The energy balance reads if take the heat capacity into account:

$$
\begin{aligned}
C_p \partial_t T &= (1-\alpha)S \cdot \cos\varphi \cdot \cos\Theta &- \epsilon\sigma T^4 &&\text{for } -\pi/2 < \Theta < \pi/2 \\
&= &-\epsilon\sigma T^4 &&\text{elsewhere}
\end{aligned} \tag{9}
$$

with $C_p$ representing the heat capacity multiplied with the depth of the atmosphere-ocean layer ($C_p$ is in the order of $10^7 - 10^8 \, JK^{-1}m^{-2}$). To simplify (9), the energy balance is integrated over the longitude and day, $\tilde{T} = \frac{1}{2\pi}\int_0^{2\pi} T \, d\Theta$. We assume that 15    the exchange of operations zonal/time mean and exponentiate to the power of 4 is valid

$$\frac{1}{2\pi}\int_0^{2\pi} T^4 \approx \tilde{T}^4 \tag{10}$$

which is a good approximation in the climate system (Fig. A.1) when analyzing hourly data temperature (Hersbach et al., 2020). Fig. A.1b indicates that the difference is lowest at low latitudes or marine-dominated regions where the diurnal variation of sea surface temperatures is small (cf., Kawai and Kawamura, 2002; Stommel, 1969; Stuart-Menteth, et al. 2003; Ward, 2006). 20    Therefore,

$$C_p \partial_t \tilde{T} = (1-\alpha)S\cos\varphi \cdot \underbrace{\frac{1}{2\pi}\int_{-\pi/2}^{\pi/2}\cos\Theta \, d\Theta}_{2} - \epsilon\sigma\underbrace{\frac{1}{2\pi}\int_0^{2\pi} T^4 \, d\Theta}_{\approx\tilde{T}^4} = (1-\alpha)\frac{S}{\pi}\cos\varphi - \epsilon\sigma\tilde{T}^4 \quad . \tag{11}$$

It is emphasized that the assumption in (10) is made for the averaging of the diurnal cycle and longitude. It is completely different from the approach in (3,5). Note furthermore that $\epsilon\sigma\tilde{T}^4$ has a pronounced latitudinal dependence, varying over two orders of magnitude. Treating this term as a constant number as in (3) is questionable.

The right hand side of equation (11) is now time independent and the equilibrium solution is simply

$$\tilde{T}(\varphi) = \sqrt[4]{\frac{4}{\pi}} \cdot \sqrt[4]{\frac{(1-\alpha)S}{4\epsilon\sigma}} \quad (\cos\varphi)^{1/4} \quad \text{shown as the red line in Fig. 3.} \tag{12}$$

The global mean temperature is

$$\overline{\tilde{T}} = \sqrt[4]{\frac{4}{\pi}} \cdot \sqrt[4]{\frac{(1-\alpha)S}{4\epsilon\sigma}} \; \underbrace{\frac{1}{2} \int\limits_{-\pi/2}^{\pi/2} (\cos\varphi)^{5/4} d\varphi}_{\sqrt{\pi}\Gamma(9/8)/\Gamma(13/8)} = \sqrt{\frac{\pi}{2}} \frac{\Gamma(9/8)}{\Gamma(13/8)} \cdot \sqrt[4]{\frac{(1-\alpha)S}{4\epsilon\sigma}} \tag{13}$$

with the factor $G = \sqrt{\frac{\pi}{2}}\frac{\Gamma(9/8)}{\Gamma(13/8)} \approx 0.989$ and $\overline{\tilde{T}} = 285$ K.

Alternatively, one can obtain the numerical solution of (9) shown as the brownish dashed line in Fig. 3 where the diurnal cycle has been explicitly taken into account. Here, $C_p = C_p^a$ has been chosen as the atmospheric heat capacity $C_p^a = c_p p_s/g = 1004\,JK^{-1}kg^{-1} \cdot 10^5 Pa/(9.81 ms^{-2}) = 1.02 \cdot 10^7\,JK^{-1}m^{-2}$ which is the specific heat at constant pressure $c_p$ times the total mass $p_s/g$. $p_s$ is the surface pressure and $g$ the gravity. The global mean temperature $\overline{T}$ is 286 K, again close to 288 K.

The effect of heat capacity is systematically analyzed in Fig. 5. The temperatures are relative insensitive for a wide range of $C_p$. We find a severe drop in temperatures for heat capacities below 0.01 of the atmospheric heat capacity $C_p^a$. Fig. 4 shows the temperature dependence for different values of $C_p$ and the length of the day, indicating a pronounced temperature drop during night for low values of heat capacities and for (hypothetical) long days of 240 h instead of 24 h. We have chosen this feature for a particular latitude (here: $45^0$N). The analysis shows that the effective heat capacity is of great importance for the temperature, this depends on the atmospheric planetary boundary layer (how well-mixed with small gradients in the vertical) and the depth of the mixed layer in the ocean which will be analyzed later.

Quite often the linearization the long wave radiation $\epsilon\sigma T^4$ is linearized in energy balance models. Indeed the linearization is performed around $0°$C (North et al., 1975a, b; Chen et al., 1995; Lohmann and Gerdes, 1998; North and Kim, 2017) and is formulated as

$$A + B \cdot T \quad . \tag{14}$$

As the temperatures based on the local energy balance without a heat capacity would vary between $T_{min} = 0$ K and $T_{max} = \sqrt{2} \cdot 288$ K $= 407$ K, a linearization would be not permitted. Fig. A.2 shows that a linear fit for the full range of these temperatures would provide high residuals (blue line) compared to the fit of observational ranges in temperature between $-50$ to $30°$C (thin red line). The thick red line takes Budyko's (1969) empirical values for present climate conditions, $A = 203.3\,Wm^{-2}$ and $B = 2.09\,Wm^{-2}\,°C^{-1}$. Therefore, the linearization implicitly assumes the above heat capacity and fast rotation arguments. If we assume a linearization, we can repeat the calculation (5) to get

$$\overline{T} = \frac{(1-\alpha)S/4 - A}{B} \tag{15}$$

with $\overline{T} = 288$ K taking Budyko's (1969) values for A and B.

## 2.3 Meridional heat transport

Equation (11) shall be the starting point for further investigations. One can easily include the meridional heat transport by diffusion which has been previously used in one-dimensional EBMs (e.g. Adem, 1965; Sellers, 1969; Budyko, 1969; North, 1975a,b). In the following we will drop the tilde sign. Using a diffusion coefficient k, the meridional heat transport across a latitude is $HT = -k\nabla T$. One can solve the EBM

$$C_p \partial_t T = -\nabla \cdot HT + (1-\alpha)\frac{S}{\pi}\cos\varphi - \epsilon\sigma T^4 \quad . \tag{16}$$

numerically. The boundary condition is that the HT at the poles vanish. The values of k are in the range of earlier studies (North, 1975a,b; Stocker et al., 1992; Chen et al., 1995; Lohmann et al., 1996). Fig. 6 shows the equilibrium solutions of (16) using different values of $k$ (solid lines). The global mean temperature is not affected by the transport term because it depends only of global net radiative fluxes, not internal redistribution. Formally, the integration with boundary condition with zero heat transport at the poles provides no effect (note that $\partial_y T = 0$ at the North and South Pole). The same is true if we introduce zonal transports because of the cyclic boundary condition in $\theta-$direction.

Until now, we assumed that the Earth's axis of rotation were vertical with respect to the path of its orbit around the Sun. Instead Earth's axis is tilted off vertical by about 23.5 degrees. As the Earth orbits the Sun, the tilt causes one hemisphere to receive more direct sunlight and to have longer days. This is a redistribution of heat with more solar insolation at the poles and less at the equator (formally it could be associated to an enhanced meridional heat transport HT). The resulting temperature is shown as the dotted blue line in Fig. 6. A spatially constant temperature in (1) can be formally seen as a system with infinite diffusion coefficient $k \to \infty$ (black line in Fig. 6).

The global mean temperatures are not affected by the tilt and the values are identical to the one calculated in (13). This is true even if we calculate the seasonal cycle (Berger and Loutre, 1991; 1997; Laepple and Lohmann, 2009). However, if we include non-linearities such as the ice-albedo feedback ($\alpha$ as a function of T), the global mean value is changing (Budyko, 1969; Sellers, 1969; North et al., 1975a, b), cf. the dashed blue line in Fig. 6. Such model can be improved by including an explicit spatial pattern with a seasonal cycle to study the long-term effects of climate to external forcing (Adem, 1981; North et al., 1983) or by adding noise mimicking the effect of short-term features on the long-term climate (Hasselmann, 1976; Lemke, 1977; Lohmann, 2018). A spatial explicit model would include the land-sea mask, heat capacities over land and the ocean, etc. This is, however, beyond the scope of the present paper. In the following we will analyze the effect of the heat capacity onto the seasonal cycle, and its effect on the global temperature.

## 2.4 Effect of the seasonal cycle

As a logical next step, let us now include an explicit seasonal cycle into the EBM:

$$C_p \partial_t T = -\nabla \cdot HT + (1-\alpha)S(\varphi,t) - \epsilon\sigma T^4 \quad . \tag{17}$$

with $S(\varphi, t)$ being calculated daily (Berger and Loutre, 1991; 1997). Eq. (17) is calculated numerically for fixed diffusion coefficient $k = 1.5 \cdot 10^6 \, m^2/s$ under present orbital conditions. Fig. 7 indicates that the temperature gradient is getting flatter for large heat capacities. Furthermore, the mean temperature is affected by the choice of $C_p$. In the case of large heat capacity at high latitudes (for latitudes polewards of $\varphi = 50°$ mimicking large mixed layer depths) and moderate elsewhere, we observe strong warming at high latitudes with moderate warming at low latitudes (dashed curve). This again indicates that we cannot neglect the time-dependent left hand side in the energy balance equations.

The question remains why the mean temperature in the dashed curve is much higher than the blue curve. Fig. 8 shows the seasonal amplitude for the $C_p$-scenarios as indicated by the blue and dashed black lines, respectively. We see the large variation in the seasonal cycle $\Delta T = T_{summer} - T_{winter}$ for the blue line in Fig. 8 as compared to the dashed line. A reduced seasonal cycle is responsible for significant warming due to $\epsilon \sigma \cdot T^4$. To estimate the order of magnitude, we can assume that the annual mean change in the net long wave radiation can be approximated by the mean of summer and winter values

$$\epsilon \sigma \cdot \overline{T^4} \quad \approx \quad \epsilon \sigma \cdot 0.5 \cdot (T_{summer}^4 + T_{winter}^4) \quad . \tag{18}$$

If the seasonal cycle is damped or in the extreme case is zero, the net long wave radiation can be approximated by

$$\epsilon \sigma \cdot \overline{T}^4 \quad \approx \quad \epsilon \sigma \cdot (0.5 \cdot [T_{summer} + T_{winter}])^4 . \tag{19}$$

A lower seasonal cycle provides for a lower net outgoing radiation, symbolically (19) < (18).

Fig. 9 shows the change in net outgoing longwave radiation with seasonal amplitude of temperature $\Delta T$ for different annual mean temperatures T. The temperature-dependence on longwave radiation is relatively minor compared to $\Delta T$. In the simulation with the larger seasonal contrast, the outgoing long wave radiation is up to $10 \, W m^{-2}$ higher, yielding a colder climate (compare the blue and dashed black lines in Fig. 7). It is noted that this feature is missing in the linearized version (14) of the outgoing radiation. Therefore, the change in seasonal amplitude does not directly influence the mean temperature in the linear version. In the following, we will analyze the change in seasonality and mean climate in a complex model.

## 2.5 Seasonal temperature changes in a complex model

A complex circulation model is used where the seasonal cycle is reduced by enhanced vertical mixing in the ocean. To make a rough estimate of the involved mixed layer, one can see that the effective heat capacity of the ocean is time-scale dependent. A diffusive heat flux goes down the gradient of temperature and the convergence of this heat flux drives a ocean temperature tendency:

$$C_p^o \partial_t T = -\partial_z (k^o \partial_z T) \tag{20}$$

where $k_v = k^o / C_p^o$ is the oceanic vertical eddy diffusivity in $m^2 \, s^{-1}$, and $C_p^o$ the oceanic heat capacity relevant on the specific time scale. The vertical eddy diffusivity $k_v$ can be estimated from climatological hydrographic data (Olbers et al., 1985; Munk and Wunsch, 1998) and varies roughly between $10^{-5}$ and $10^{-4} \, m^2 \, s^{-1}$ depending on depth and region.

A scale analysis of (20) yields a characteristic depth scale $h_T$ through

$$\frac{\Delta T}{\Delta t} = k_v \frac{\Delta T}{h_T^2} \quad \longrightarrow h_T = \sqrt{k_v \, \Delta t} \tag{21}$$

For the diurnal cycle $h_T$ is less that half a meter and the heat capacity generally less than that of the atmosphere. The seasonal mixed layer depth can be several hundred meters (e.g., de Boyer Montégut et al., 2004). As pointed out by Schwartz (2007), the effective heat capacity that reflects only that portion of the global heat capacity that is coupled to the perturbation on the timescale of the perturbation. As an example, the effective heat capacity is subject to change in heat content in the context of global climate change induced by changes in gaseous components of the atmosphere on decadal to centennial timescales.

In order to test the effective heat capacity/mixing hypothesis, we employ the coupled climate model COSMOS which was mainly developed at the Max-Planck Institute for Meteorology in Hamburg (Jungclaus et al., 2000). The model contains explicit diurnal and seasonal cycles, it has no flux correction and has been successfully applied to test a variety of paleoclimate hypotheses, ranging from the Miocene climate (Knorr et al., 2011; Knorr and Lohmann, 2014; Stein et al., 2016), the Pliocene (Stepanek and Lohmann, 2012) as well as glacial (Zhang et al., 2013; 2014) and interglacial climates (Wei and Lohmann, 2012; Lohmann et al., 2013; Pfeiffer and Lohmann, 2016).

In order to mimic the effect of a higher effective heat capacity and deepened mixed layer depth, the vertical mixing coefficient is increased in the ocean, changing the values for the background vertical diffusivity (arbitrarily) by a factor of 25, providing a deeper thermocline. The mixing has a background value plus a mixing process strongly influenced by the shears of the mean currents. Although observations give a range of values of $k_v$ for the ocean interior, models use simplified physics and prescribe a constant background value. The model uses a classical vertical eddy viscosity and diffusion scheme (Pacanowski and Philander, 1981). Orbital parameters are fixed to the present condition.

Fig. 10 shows the anomalous near surface temperature for the new vertical mixing experiment relative to the control climate (Wei and Lohmann, 2012). Both simulations were run over 1000 years of integration in order to receive a quasi-equilibrium at the surface. The differences are related to the last 100 years of the simulation. In the vertical mixing experiment $k_v$ was enhanced leading to more heat is taken up by the ocean producing equable climates with pronounced warming at polar latitudes (Fig. 10). Heat gained at the surface is diffused down the water column, and, compared to the control simulation, the wind-induced Ekman cells in the upper part of the oceans intensified and deepened. The model indicates that the respective winter signal of high-latitude warming is most pronounced (Fig. 10), decreasing the seasonality, suggesting a common signal of pronounced warming and weaker seasonality, a feature already seen in the EBM (Figs. 8, 9).

## 3 Discussion

We can discuss the assumptions for obtaining global mean surface temperature of the Earth. We get realistic temperatures for the following cases

– Equation (3) where we treat the temperature on the Earth as one number. This approach is written in most text books of climate. The averaging is problematic since $T^4$ has a pronounced latitudinal dependence. Furthermore, the solution does not take the heat capacity and the Earth's rotation rate into account.

– Equation (13) where we assume a significant heat capacity and fast rotation of the Earth. The diurnal cycle is averaged out. The global mean temperature is similar to a (3), but multiplied with a factor $G = \sqrt{\frac{\pi}{2}} \frac{\Gamma(9/8)}{\Gamma(13/8)} \approx 0.989$

- One can obtain the numerical solution of (9) where the diurnal cycle has been explicitly taken into account. The global mean temperature is similar to the previous solution and close to observations.

- If we linearize the outgoing radiation (14) which implicitly assumes the above heat capacity and fast rotation arguments, we get a realistic $\overline{T} = 288$ K for present climate conditions (15) .

- The global mean temperatures in slightly more complex EBMs (16,17) are in the range of observations for significant effective heat capacities.

- The global mean temperature is realistically simulated in a complex GCM with diurnal and seasonal cycle, as e.g. used in subsection 2.5 (Stepanek and Lohmann, 2012; Wei and Lohmann, 2012).

In the EBM it is found that a low effective heat capacity enhances the daily/seasonal cycle providing a higher outgoing longwave radiation cooling the Earth. In case of a zero heat capacity, the mean temperature is a factor $F = 0.4\sqrt{2} \approx 0.566$ lower than 288 K as stated in (3) and would be $\overline{T} = 163$ K.

We can be put our finding into a general statement. Let us define here $^{-}$ as the averaging over an arbitrary time period (in our context the seasonal/diurnal cycle), then $\overline{T^4} > \overline{T}^4$ which is consistent with Hölder's inequality (Rodgers, 1888; Hölder 1889; Hardy et al., 1934, Kuptsov, 2001). For the diurnal cycle, $\overline{T^4}$ in (5) is greater than $\overline{T}^4$ as obtained from (8). We find that $\overline{T}$ is a factor $F = 0.4\sqrt{2}$ lower than the fourth root of $\overline{T^4}$. For the seasonal cycle, the climate with the lower seasonal cycle (dashed black line in Figs. 7, 8) is warmer than state with a more pronounced seasonal cycle (blue line in Figs. 7, 8). This is due to the strong dependence of the outgoing radiation onto the seasonal range of temperatures (Fig. 9). The higher the daily/seasonal contrast, the outgoing longwave radiation is increased. We examine that this is related to the effective heat capacity of the system.

The effective heat capacity is connected to the perturbation on the timescale of the perturbation (21). Previous studies have noted that changing the ocean mixed layer depth impacts the climatological annual mean temperature (Schneider and Zhu, 1998; Qiao et al., 2004; Donohoe et al., 2014; Wang et al., 2019). The increased heat capacity of the mixed layer reduced the magnitude of the annual cycle affecting the surface winds and upwelling which may provide non-linear effects (Wang et al., 2019). For the past, a strong warming at high latitudes is reconstructed for the Pliocene, Miocene, Eocene periods (Markwick, 1994; Wolfe, 1994; Sloan and Rea, 1996; Huber et al., 2000; Shellito et al., 2003; Tripati et al., 2003; Mosbrugger et al., 2005; Utescher and Mosbrugger, 2007). It is a conundrum that the modelled high latitudes are not as warm as the reconstructions (e.g., Sloan and Rea, 1996; Huber et al., 2000; Mosbrugger et al., 2005; Knorr et al., 2011; Dowset et al., 2013). Inspired by the EBM and GCM results, we may think of a climate system having a higher effective heat capacity producing a reduced seasonal cycle and flat temperature gradients. The changed vertical mixing coefficients could mimick possible effects like weak tidal dissipation or abyssal stratification (e.g., Lambeck 1977; Green and Huber, 2013), but its explicit physics is not evaluated here.

## 4  Conclusions

This manuscript revisits the relationship between the (global mean) surface temperature of the Earth and its radiation budget as is frequently used in Energy balance models (EBMs). The main point is, that the effective heat capacity and its temporal variation over the daily/seasonal cycle needs to be taken into account when estimating surface temperature from the energy budget. EMBs provide a crucial tool in climate research, especially because they - confirmed by the results of the elaborate realistic climate models - describe the processes essential for the genesis of the global climate. EBMs are thus an admissible conceptual tools, due to its reduced complexity to the essentials "scientific understanding" represents (von Storch et al., 1999). This understanding states that the radiation balance on the ground and the absorption in the atmosphere are the essential factors for determining the temperature. The solution of the basics EBM says that the temperature is independent of the size of the Earth and the thermal characteristics, but depends on the albedo, emissivity and solar constant.

The argument follows the conservation of energy: in steady state the Earth has to emit as much energy as it receives from the Sun. However, I argue that we shall explicitly emphasize the Earth as a rapidly rotating object with a significant heat capacity. Without these effects, the global mean temperature would be much lower and can be better used for objects like the Moon or Mercury (Vasavada et al., 1999; Lorenz, 2005) as slowly rotating bodies without significant heat capacity. The Earth system understanding says that the effect of the effective heat capacity is important for the radiation balance, other processes - like horizontal transport processes or the ice-albedo feedback - are only of secondary importance for the globally averaged temperature. Interestingly, the global mean temperature in the revised EBM is very close to the original proposed value.

As a basic feature, the net outgoing longwave radiation is reduced if the diurnal or seasonal cycle is reduced, providing for a significant warming. On long time scales, the effective heat capacity is linked to the mixed-layer depth of the ocean. A change in the mixed layer depth which likely happened through glacial-interglacial cycles (e.g. Zhang et al., 2014) can therefore an important driver constraining climate sensitivity (Köhler, et al., 2010). This could be also relevant for past and future climate change when the ocean stratification and mixing can change. The temperature-dependence is indeed emphasized in a sensitivity study of climatological SST to slab ocean model thickness (Wang et al., 2019). It might be that the more effective mixing provides an explanation that high latitudes were much warmer than present and more equable in that the summer-to-winter range of temperature was much reduced (Sloan and Barron, 1990, Valdes et al., 1996; Sloan et al., 2001; Spicer et al. 2004). Interestingly, it has been suggested that the tight link between ocean temperature and $CO_2$ formed only during the Pliocene when the thermocline shoals and surface water became more sensitive to $CO_2$ (La Riviere et al., 2012) which is therefore of major importance for the understanding of the climate-carbon cycle (Wiebe and Weaver, 1999; Zachos et al., 2008; de Boer and Hogg, 2014).

It is concluded that climate studies should use improved representations of vertical mixing processes including turbulence, tidal mixing, hurricanes and wave breaking (e.g., Qiao et al., 2004; Huber et al., 2004; Simmons et al., 2004; Korty et al., 2008; Griffiths and Peltier, 2009; Green and Huber, 2013; Reichl and Hallberg, 2018). Global climate models treat ocean vertical mixing as static, although there is little reason to suspect this is correct (e.g., see Munk and Wunsch, 1998). In numerical modelling, the values are also constrained by the required numerical stability and to fill gaps left by other parameterisations

(e.g., Griffies, 2005). As a natural next step, one shall analyze the ocean mixing/heat uptake (Luyten et al., 1983; Large et al., 1994; Nilsson, 1995) to understand past, present and future temperatures.

*Acknowledgements.* Thanks go to Peter Köhler, Dirk Olbers, anonymous referees, and the editor for comments on earlier versions of the manuscript. Madlene Pfeiffer and Christian Stepanek are acknowledged for their contribution in producing Fig. 10 and Fig. A.1. Reanal-
5  ysis data used in this analysis were provided by the Copernicus Climate Change Service (Hersbach et al. 2020), the European Centre for Medium-Range Weather Forecasts (https://apps.ecmwf.int/datasets/data/interim-full-daily/levtype=pl/). This work was funded by the Helmholtz Society through the research program PACES.

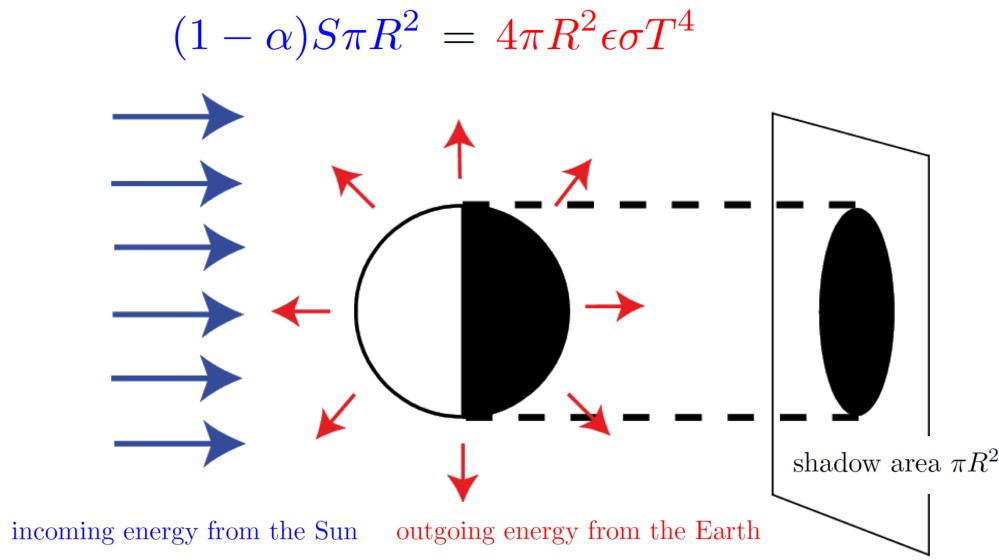

**Figure 1.** Schematic view of the energy absorbed and emitted by the Earth following (1). Modified after Goose (2015).

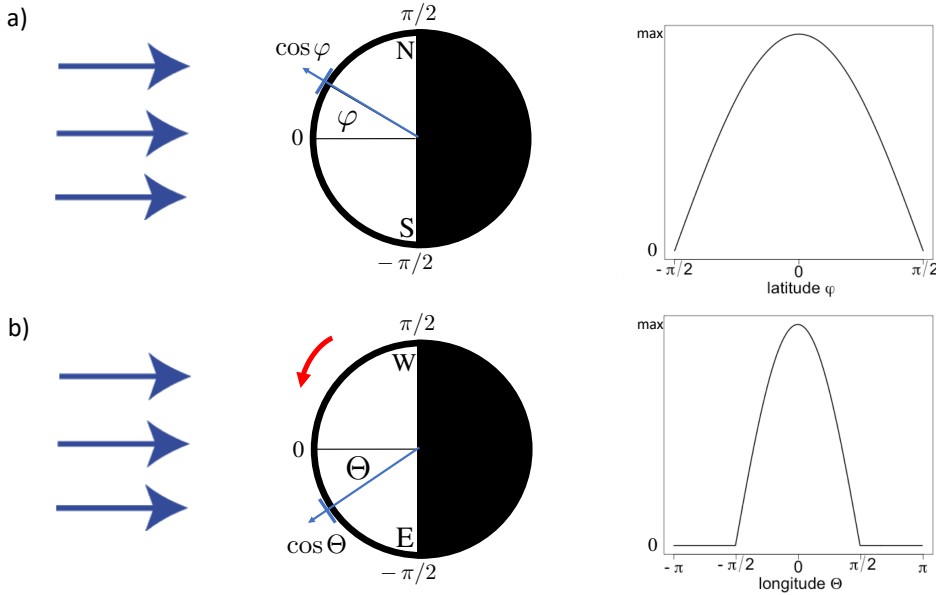

**Figure 2.** Latitudinal (a) and longitudinal (b) dependence of the incoming short wave radiation. On the right hand side, the insolation as a function of latitude $\varphi$ and longitude $\Theta$ with maximum insolation $(1-\alpha)S$ is shown. See text for the details.

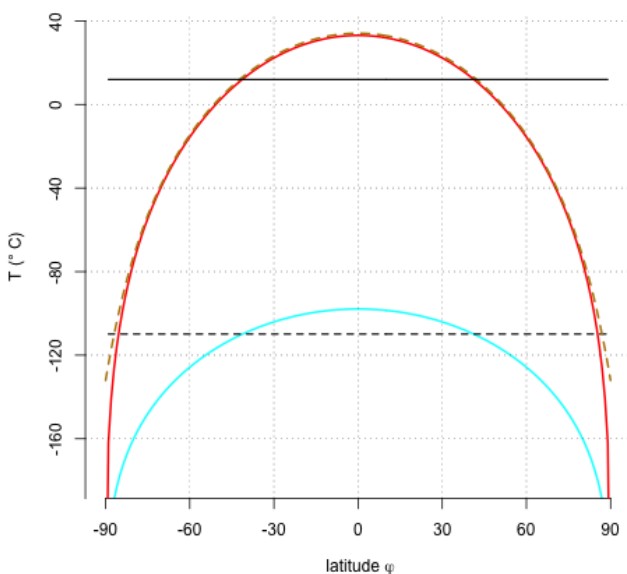

**Figure 3.** Latitudinal temperatures of the EBM with zero heat capacity (7) in cyan (its mean as a dashed line), the global approach (3) as solid black line, and the zonal and time averaging (12) in red. Please note the both curves are exactly $0K = -273.15°\text{C}$ at the poles but those values were not shown for graphical reasons. The dashed brownish curve shows the numerical solution by taking the zonal mean of (9).

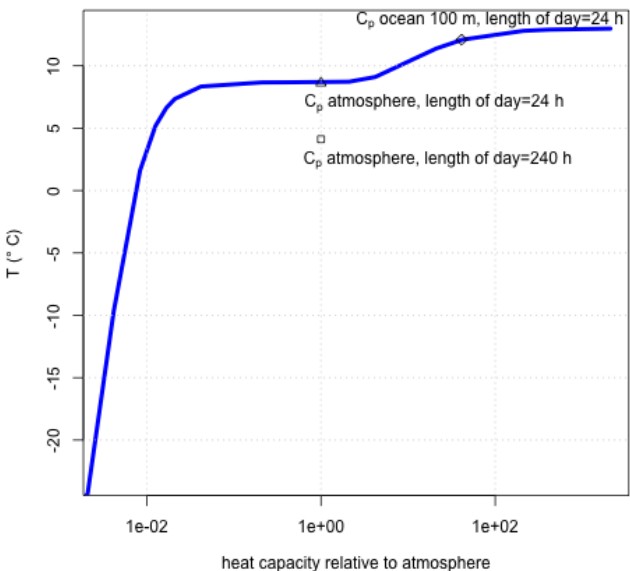

**Figure 4.** Temperature dependence on heat capacity (and rotation rate) when analyzing the daily mean temperature at 45°N using (9).

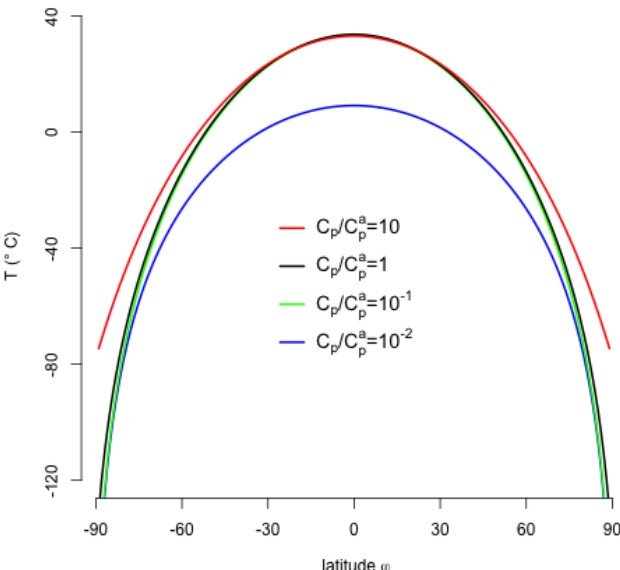

**Figure 5.** Temperature depending on $C_p$ when solving (9) numerically. The reference heat capacity is the atmospheric heat capacity $C_p^a = 1.02 \cdot 10^7\, JK^{-1}m^{-2}$. The climate is insensitive to changes in heat capacity $C_p \in [0.05 \cdot C_p^a, 2 \cdot C_p^a]$.

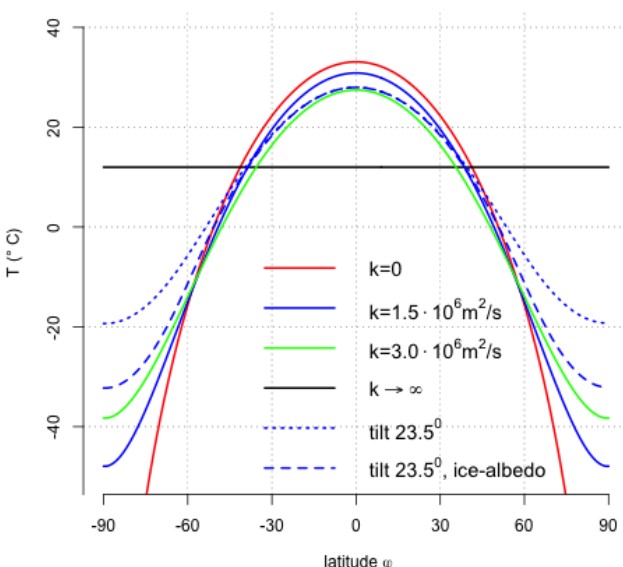

**Figure 6.** Equilibrium temperature of (16) using different diffusion coefficients. $C_p = C_p^a$. The blue lines use $1.5 \cdot 10^6 m^2/s$ with no tilt (solid line), a tilt of $23.5°$ (dotted line), and as the dashed line a tilt of $23.5°$ (present value) and ice-albedo feedback using the respresentation of Sellers (1969). Except for the dashed line, the global mean values are identical to the value calculated in (13). Units are $°$C.

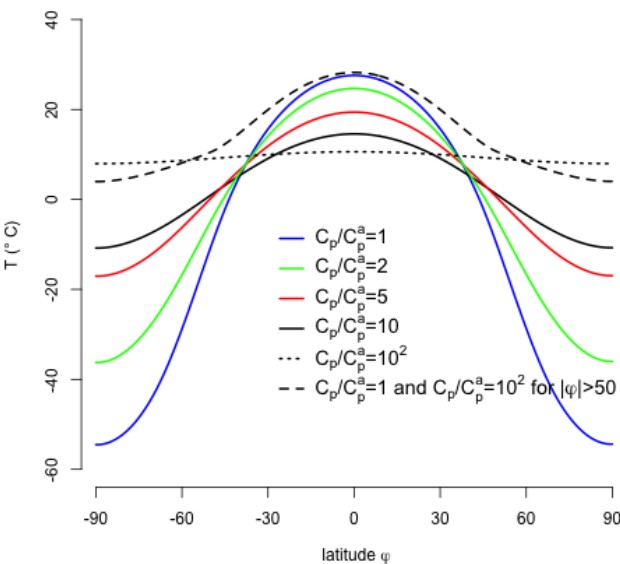

**Figure 7.** Annual mean temperature depending on $C_p$ when solving the seasonal resolved EBM (17) numerically. For all solutions, we use use $k = 1.5 \cdot 10^6 m^2/s$, present day orbital parameters, and the ice-albedo feedback using the respresentation of Sellers (1969).

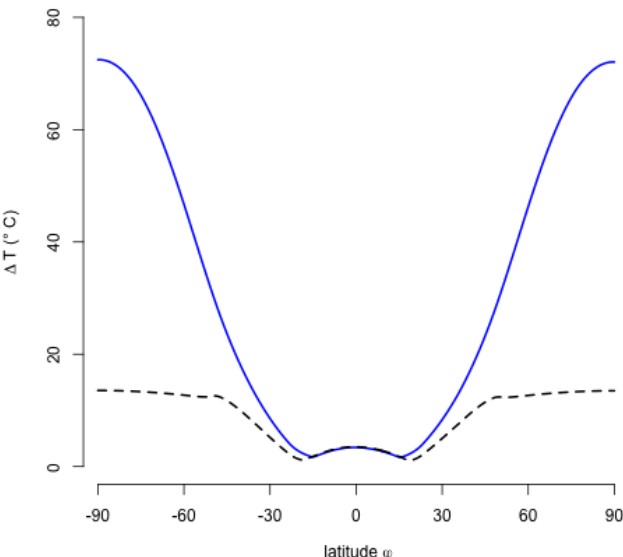

**Figure 8.** Seasonal amplitude of temperature for the two extreme scenarios in Fig. 7, indicating that a lower seasonality as in the dashed-black line relative to the blue line is linked to a warmer annual mean climate.

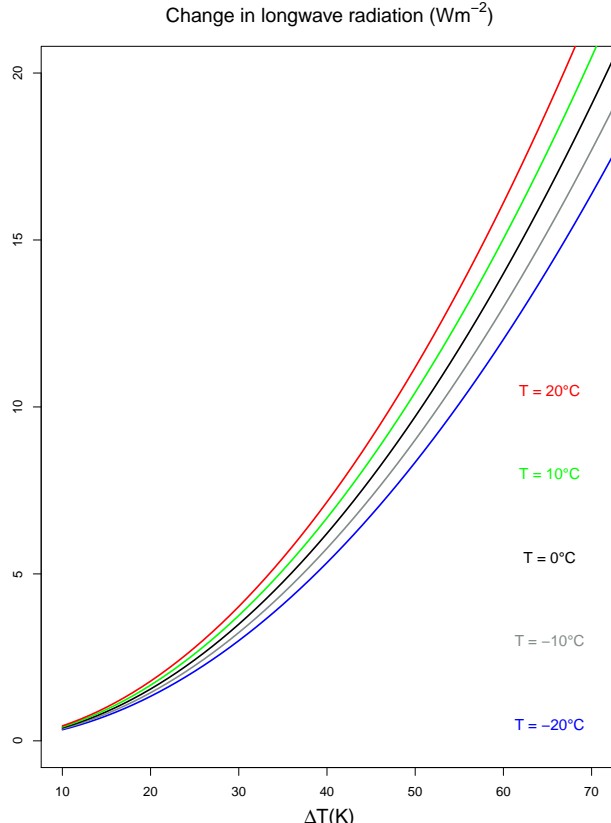

**Figure 9.** Longwave radiation change with seasonal amplitude of temperature $\Delta T$ for different temperatures T.

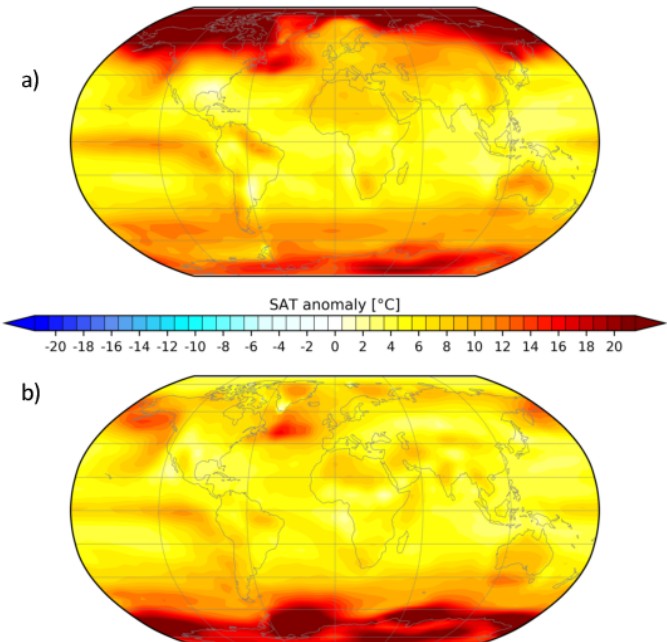

**Figure 10.** Anomalous near surface temperature for the vertical mixing experiment relative to the control climate. a) Mean over boreal winter and austral summer (DJF), b) Mean over austral winter and boreal summer (JJA). Shown is the 100 year mean after 900 years of integration using the Earth system model COSMOS. Units are °C.

## Appendix A

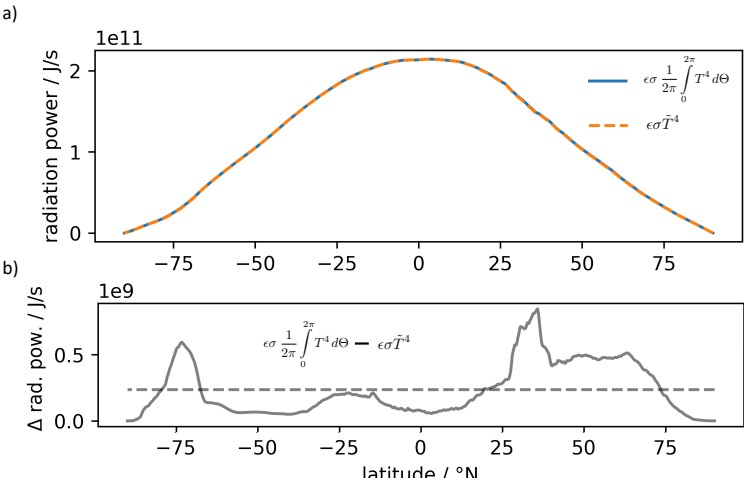

**Figure A.1.** Longwave radiation $\epsilon\sigma T^4$ based on hourly near surface temperatures, generated using Copernicus Climate Change Service information for 1979 AD (doi:10.24381/cds.adbb2d47, 2018-06-14), Hersbach et al. (2020). a) It is seen that $\epsilon\sigma \frac{1}{2\pi} \int_0^{2\pi} T^4 \, d\Theta \approx \epsilon\sigma \tilde{T}^4$ . b) The differences are more than two orders of magnitude smaller than the mean values. The most pronounced differences are over the subtropics (deserts) and mid-to-high latitudes where we have a pronounced daily cycle.

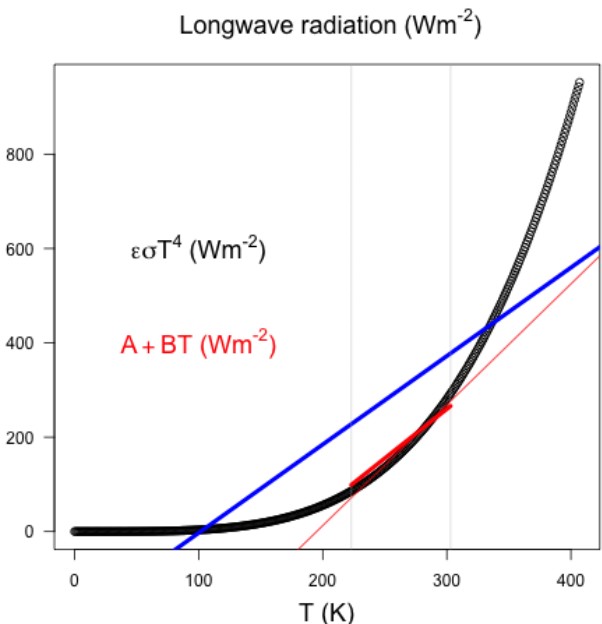

**Figure A.2.** Black dots: $\epsilon\sigma T^4$, blue line: linear fit for the range of temperatures 0 to 407 K. Thin red line: Linear fit for temperature between $-50$ to $30°$C (range is shown as the vertical dotted lines). Thick red line shows Budyko's (1969) linearization with $A = 203.3\,Wm^{-2}$ and $B = 2.09\,Wm^{-2}\,°C^{-1}$. The regression coefficients for the blue line is $A = 321.4\,Wm^{-2}, B = 1.88\,Wm^{-2}\,°C^{-1}$, whereas for the the thin red line is $A = 199.5\,Wm^{-2}, B = 2.56\,Wm^{-2}\,°C^{-1}$.

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
