# Peer review of "Temperatures from Energy Balance Models: the effective heat capacity matters"

_Earth System Dynamics, 2019_

## Referee Comment (RC1) · Anonymous Referee #1 · 4 Oct 2019

This manuscript revisits the relationship between the (global mean) surface temperature of the Earth and its radiation budget as is frequently used in Energy balance models (EBMs). The main point is, that the effective heat capacity (and its temporal variation over the daily/seasonal cycle) needs to be taken into account when estimating surface temperature from the energy budget. The results of this exercise together with coupled ocean-atmosphere GCM simulations lets the author suggest a potential mechanism for the relatively low equator-to-pole temperature gradient in past warm climates that has been observed in proxy data, but remains difficult to reproduce with GCMs.

The paper includes a very useful discussion about general properties of the energy balance of the Earth and this certainly justifies publication in ESD. However, I have two

main comments to be improved on before I can recommend publication:

1. The theoretical arguments should be much better explained. This holds in particular for sections 2 and 3.

   For example, after or before eq. (4), it should be very explicitly explained which variables become lat-lon dependent, and which not. Otherwise eq. (4) and the analysis that follows is very hard to understand (or reproduce). In my view, if you consider the local energy balance, temperature $T$, emissivity $\epsilon$ and albedo $\alpha$, should be spatially dependent and therefore this should have consequences for the following integration. If they are not spatially dependent, then it should be clearly stated why not.

   I find it very puzzling that the heat capacity $C_p$ does not explicitly appear in eq. (11), although I clearly see how you get there. A few words of explanation would be very useful to the (less-expert) reader.

   Then, after eq. (12) the reference heat capacity is chosen as the atmospheric heat capacity. Why is that? Above in the text you have said that the heat capacity is manly given by the ocean, so why do you use the atmospheric heat capacity here?

   A bit more explanation and motivation should also enter the fact that in one case in Fig. 5 you use a latitudinal dependent heat capacity (in the text just after eq. (12)). How exactly? And what is the motivation for that?

   On page 6, line 18, the temperatures T1 and T2 remain unexplained!

2. The second point relates to the vertical mixing in the ocean. It is interesting to see how the vertical mixing in the ocean obviously can affect the equator-to-pole surface temperature gradient. However, why should the vertical mixing be so different in the Palaeogene/Neogene before 3 Ma? Tidal dissipation can play a role, but also bathymetry and probably also the number and specific geometry of

the ocean gateways. But so far, this remains very speculative and unmotivated in the manuscript. For example, how does the factor 25 in the vertical mixing coefficient that is used in the GCM simulations relate to expected changes in vertical mixing due to tides and bathymetry?

---

## Author Comment (AC1) · 8 Oct 2019

Thanks for the constructive critics in the *Interactive comment on Earth Syst. Dynam. Discuss., https://doi.org/10.5194/esd-2019-35, 2019.* by referee #1. In the following, I will repeat and answer to these comments. Furthermore, possible action is proposed.

**Comment**

This manuscript revisits the relationship between the (global mean) surface temperature of the Earth and its radiation budget as is frequently used in Energy balance models (EBMs). The main point is, that the effective heat capacity (and its temporal variation over the daily/seasonal cycle) needs to be taken into account when estimating surface temperature from the energy budget. The results of this exercise together with coupled ocean-atmosphere GCM simulations lets the author suggest a potential mechanism for the relatively low equator-to-pole temperature gradient in past warm climates that has been observed in proxy data, but remains difficult to reproduce with GCMs. The paper includes a very useful discussion about general properties of the energy balance of the Earth and this certainly justifies publication in ESD. However, I have two main comments to be improved on before I can recommend publication.

**Comment 1a)**

The theoretical arguments should be much better explained. This holds in particular for sections 2 and 3. For example, after or before eq. (4), it should be very explicitly explained which variables become lat-lon dependent, and which not. Otherwise eq. (4) and the analysis that follows is very hard to understand (or reproduce). In my view, if you consider the local energy balance, temperature T, emissivity and albedo $\alpha$, should be spatially dependent and therefore this should have consequences for the following integration. If they are not spatially dependent, then it should be clearly stated why not.

**Answer/Action**

Indeed, eq. (4) can be better explained. The left hand side of (4) as well as the right hand side are latitude $\varphi$- and longitute $\Theta$-dependent.

$$\epsilon \sigma T^4(\Theta, \varphi) \quad = \quad (1-\alpha)S \quad \cos\varphi \quad \cos\Theta \quad \times \quad 1_{[-\pi/2<\Theta<\pi/2]}(\Theta)$$

The incoming radiation goes with the cosinus of latitude and longitude, and there is only sunshine during the day. This is noted as the $1_{[-\pi/2<\Theta<\pi/2]}(\Theta)$ function which is zero outside the interval $[-\pi/2 < \Theta < \pi/2]$. The beauty of this formulation is that we ignore the Earth orbital parameters for a while which makes an analytical calculation possible. A temperature-dependent formulation of the albedo is used later because of simplicity ($\alpha$ as a function of T).

If we now calculate the zonal mean of the temperature by integration at the latitudinal cycles we have

$$
\begin{aligned}
T(\varphi) &= \frac{1}{2\pi} \int_{-\pi/2}^{-\pi/2} \sqrt[4]{\frac{(1-\alpha)S\cos\varphi\cos\Theta}{\epsilon\sigma}}\,d\Theta \\[2mm]
&= \frac{\sqrt{2}}{2\pi} \underbrace{\int_{-\pi/2}^{\pi/2} (\cos\Theta)^{1/4}d\Theta}_{=\sqrt{\pi}\Gamma(5/8)/\Gamma(9/8)} \quad \sqrt[4]{\frac{(1-\alpha)S}{4\epsilon\sigma}} \;(\cos\varphi)^{1/4} \\[2mm]
&= \frac{1}{\sqrt{2\pi}} \frac{\Gamma(5/8)}{\Gamma(9/8)} \cdot \sqrt[4]{\frac{(1-\alpha)S}{4\epsilon\sigma}} \;(\cos\varphi)^{1/4} \\[2mm]
&= 0.608 \cdot \sqrt[4]{\frac{(1-\alpha)S}{4\epsilon\sigma}} \;(\cos\varphi)^{1/4}
\end{aligned}
\tag{1}
$$

as a function on latitude. $\Gamma$ is Euler's Gamma function with $\Gamma(x+1) = x\Gamma(x)$. When we integrate this over the latitudes, we obtain

$$
\begin{aligned}
\overline{T} &= \frac{1}{2} \int_{-\pi/2}^{\pi/2} T(\varphi)\cos\varphi\,d\varphi \\[2mm]
&= \frac{1}{2} \frac{\Gamma(5/8)}{\sqrt{2\pi}\,\Gamma(9/8)} \cdot \sqrt[4]{\frac{(1-\alpha)S}{4\epsilon\sigma}} \quad \underbrace{\int_{-\pi/2}^{\pi/2} (\cos\varphi)^{5/4}d\varphi}_{=\sqrt{\pi}\Gamma(9/8)/\Gamma(13/8)}
\end{aligned}
$$

$$
\begin{aligned}
&= \frac{1}{2} \frac{1}{\sqrt{2}} \frac{\Gamma(5/8)}{\Gamma(13/8)} \sqrt[4]{\frac{(1-\alpha)S}{4\epsilon\sigma}} = \frac{\sqrt{2}}{4} \frac{8}{5} \sqrt[4]{\frac{(1-\alpha)S}{4\epsilon\sigma}} \\[2mm]
&= 0.4\sqrt{2} \sqrt[4]{\frac{(1-\alpha)S}{4\epsilon\sigma}} = 0.566 \sqrt[4]{\frac{(1-\alpha)S}{4\epsilon\sigma}}
\end{aligned}
\tag{2}
$$

**Comment 1b)**

I find it very puzzling that the heat capacity $C_p$ does not explicitly appear in eq. (11), although I clearly see how you get there. A few words of explanation would be very useful to the (less-expert) reader.

**Answer/Action**

Indeed, the equilibrium solution is calculated from (10) as there is no time-dependence due to the integration over longitude and day. I will be more explicit in the revised version in explaining the $1_{[-\pi/2<\Theta<\pi/2]}(\Theta)$ function.

**Comment 1c)**

Then, after eq. (12) the reference heat capacity is chosen as the atmospheric heat capacity. Why is that? Above in the text you have said that the heat capacity is manly given by the ocean, so why do you use the atmospheric heat capacity here?

**Answer/Action**

Yes, the effective heat capacity is time-scale dependent. For the day and night cycle values in the order of the atmospheric heat capacity are realistic for our Earth with 24

h rotation. In the revised version, I will be more explicit here and could include a figure to show how the temperature would change in a slowly rotating planet.

**Comment 1d)**

A bit more explanation and motivation should also enter the fact that in one case in Fig. 5 you use a latitudinal dependent heat capacity (in the text just after eq. (12)). How exactly? And what is the motivation for that?

**Answer/Action**

Yes, this can be better motivated. The high latitudes have a much higher effective heat capacity due to the deeper mixed layer.

**Comment 1e)**

On page 6, line 18, the temperatures T1 and T2 remain unexplained!

**Answer/Action**

In the revised version, these temperatures are better explained. It is a back-on-the-envelope calculation to see the main argument.

**Comment 2)**

The second point relates to the vertical mixing in the ocean. It is interesting to see how the vertical mixing in the ocean obviously can affect the equator-to-pole surface temperature gradient. However, why should the vertical mixing be so different in the Palaeogene/Neogene before 3 Ma? Tidal dissipation can play a role, but also bathymetry and probably also the number and specific geometry of the ocean gateways. But so far, this remains very speculative and unmotivated in the manuscript. For example, how does the factor 25 in the vertical mixing coefficient that is used in the GCM simulations relate to expected changes in vertical mixing due to tides and bathymetry?

**Answer/Action**

The manuscript is admittedly a little vague at this point. A more explicit statement about a more explicit calculation of the vertical mixing is beyond the scope of the present paper. I will add more literature dealing with bathymetry, tides, and geometry of the ocean in the revised version. stressed out that The effective heat capacity is not an intrinsic property of the climate system but is reflective of the rate of penetration of heat energy into the ocean in response to the particular pattern of forcing and the background state (Schwartz, 2007). I will also explore the relevance for climate warming scenarios.

---

## Referee Comment (RC2) · Anonymous Referee #2 · 24 Nov 2019

**Review**

Title: Temperature from energy balance models: the effective heat capacity
matters
Author: Gerrit Lohmann

**General Comments**

The author attempted to investigate the energy balance in terms of temperature
on the surface of the earth. He concluded that effective heat capacity is essential
in explaining the observed temperature. Unfortunately, an essential process of
heat distribution is not accounted for in the discussion, resulting in a faulty
conclusion. It is unfortunate that I cannot recommend the publication of this
paper in its present form.

**Specific Comments**

**1**. Eq. (4): $\varepsilon\sigma T^4(\varphi,\Theta)=(1-\alpha)S\cos\varphi\cos\Theta\times 1_{[-\pi/2<\Theta<\pi/2]}(\Theta)$ is not a good
description of the energy balance on the surface of the earth. A description of
the spatial distribution of energy requires an introduction of energy
redistribution by ocean currents, eddies, etc. Further, sun's declination angle
should be taken into account in order to describe reasonable spatial distribution
of energy at any specific time of the year. What is described here is, at best, an
energy balance in an annual-mean sense when there is no physical mechanism
for redistribution of strong energy surplus in the equatorial region and strong
energy deficit in the polar region.

**2**. P3 Eq. (8): This is a strange derivation. Let us consider outgoing longwave
radiation and incoming solar radiation in the form

$$\varepsilon\sigma T^4(\phi,\theta)=\left(1-\alpha\right)S\cos\phi\cos\theta\times I_{[-\pi/2<\phi<\pi/2]}(\phi), \tag{1}$$

where $\phi$ is longitude and $\theta$ is latitude. Equation (1) defines energy per unit
time per unit area as the dimension of $\sigma=5.670373\times10^{-8}$ W m$^{-2}$ K$^{-4}$ indicates.
Thus, total incoming radiation can be written as

$$\int_{-\pi/2}^{\pi/2}\int_{-\pi}^{\pi}\left(1-\alpha\right)S\cos\phi\cos\theta\times I_{[-\pi/2<\phi<\pi/2]}(\phi)R\cos\theta\,d\phi R\,d\theta$$

$$=\left(1-\alpha\right)SR^2\int_{-\pi/2}^{\pi/2}\int_{-\pi}^{\pi}\cos\phi\cos^2\theta\,d\phi\,d\theta$$

$$=\left(1-\alpha\right)SR^2\int_{-\pi/2}^{\pi/2}2\cos^2\theta\,d\theta=\left(1-\alpha\right)SR^2\int_{-\pi/2}^{\pi/2}\left(1+\cos2\theta\right)d\theta$$

$$=\left(1-\alpha\right)SR^2\int_{-\pi/2}^{\pi/2}2\cos^2\theta\,d\theta=\left(1-\alpha\right)S\pi R^2. \tag{2}$$

Similarly, total outgoing radiation can be written as

$$\int_{-\pi/2}^{\pi/2}\int_{-\pi}^{\pi}\varepsilon\sigma T^4(\phi,\theta)R\cos\theta\,d\phi\,R\,d\theta$$

$$=\varepsilon\sigma\overline{T}^4R^2\int_{-\pi/2}^{\pi/2}\int_{-\pi}^{\pi}\cos\theta\,d\phi\,d\theta=\varepsilon\sigma\overline{T}^4 4\pi R^2\,,\tag{3}$$

where

$$\overline{T}^4=\left.\int_{-\pi/2}^{\pi/2}\int_{-\pi}^{\pi}T^4(\phi,\theta)\cos\theta\,d\phi\,d\theta\middle/\int_{-\pi/2}^{\pi/2}\int_{-\pi}^{\pi}\cos\theta\,d\phi\,d\theta\right..\tag{4}$$

Thus, we arrive at

$$\overline{T}=\sqrt[4]{\frac{(1-\alpha)S}{4\varepsilon\sigma}}\,.\tag{5}$$

As stated in my Comment #3, Eq. (1) is not quite correct since it lacks the heat redistribution mechanism. On the other hand, global averaging of heat redistribution should be zero, since there is no source or sink of heat. Thus, the addition of heat redistribution to Eq. (1) does not change the result addressed above. See also my Comment #3.

**3**. P3 L14: Specific heat is not needed to reproduce the reasonable spatial distribution of temperature. For example, 1D energy balance model with meridional heat flux can be written as (see North and Kim, 2017; p123-134)

$$-\frac{d}{d\mu}\left(D(1-\mu^2)\frac{dT(\mu)}{d\mu}\right)+A+BT(\mu,t)=Qa(\mu)s(\mu)\,,\tag{6}$$

where divergence of heat flux is approximated in the form of a diffusive heat transport as $\nabla\cdot\vec{q}_{\text{heat}}=-\nabla\cdot\left(D\nabla T\right)$, the outgoing longwave radiation is linearized as $\varepsilon\sigma T^4\approx A+B\left(T-273.15\right)$, $a(\mu)=1-\alpha(\mu)$ is colatitude, $s(\mu)$ is insolation distribution function (in more general form than the author used in his Eq. (4)), and $\mu=\sin\theta=\cos\vartheta$ (sine of latitude = cosine of colatitude). Solving (1) with a solution in the form

$$T(\mu)\approx T_0+T_2 P(\mu)\tag{7}$$

with a realistic insolation distribution function and a realistic albedo (see North and Kim, 2017 for details), we obtain a solution as in Figure 1. The model solution is fairly similar to the observational data. Further, the diffusive heat transport in a zonal mean sense looks very reasonable compared to that derived from satellite observations (see Figure 2).

[Figure]

**Figure 1**. Illustration of the level of agreement at large-scale of the two-mode EBM with the observations. The solid line denotes the pole-to-pole solution of the two-model model-computed temperature (K) versus $\mu$, where $\mu \equiv \cos\vartheta$ = sine(latitude). The dashed curve indicates zonally averaged Northern Hemisphere surface air temperature taken from data in Hartmann (1994). The black-dashed curve shows the temperature curve with the $T_4$ mode added. (copied from North and Kim, 2017)

[Figure]

**Figure 2**. Illustration of the level of agreement in large-scale total poleward transport of heat from pole to pole for the two-mode EBM despite the huge differences in the geography of the two hemispheres. Poleward transport of heat derived from radiation budget data of Trenberth and Caron (2001) based on 4 years of ERBE data (1984-1988). The dashed curve represents the transport derived from the satellite observations and the solid line is based on the two-mode approximation to the surface temperature field with diffusive transport. (copied from North and Kim, 2017)

**4**. P4 L4: "The atmospheric circulation provides an efficient way to propagate heat along latitudes which is ignored and is a second order effect (not shown)." This statement is erroneous. As demonstrated in Comment #3 above, a reasonable temperature distribution on the surface of the earth is reproduced by using diffusive heat transport. Heat capacity is not even used in this calculation of equilibrium temperature.

**5**. Eq. (12): The author introduced diurnal cycle of temperature and determined the global average of the averaged diurnal cycle of temperature. This discussion is erroneous. We can write diurnal temperature change as

$$T(\phi,\theta,t) = T_0(\phi,\theta) + T_1(\phi,\theta)\cos\left(\frac{2\pi t}{24} + \varphi_1\right) + T_2(\phi,\theta)\cos\left(\frac{4\pi t}{24} + \varphi_2\right) + \cdots, \quad (8)$$

where $T_1(\phi,\theta)$ and $T_2(\phi,\theta)$ are respectively the amplitude of the diurnal cycle and of the semi-diurnal cycle with phase $\varphi_1$ and $\varphi_2$. If we average (8) over the period of 1 day, we have

$$\frac{1}{T_{\text{day}}} \int_0^{T_{\text{day}}} T(\phi,\theta,t)dt = T_0(\phi,\theta). \quad (9)$$

Further, incoming solar radiation has the same form as in (1). Thus, we arrive at the same conclusion as in Comment #2.

**6**. P4 L11: A time-dependent 1D EBM can be written as

$$C(\mu,t)\frac{\partial T(\mu,t)}{\partial t} - \frac{\partial}{\partial \mu}\left(D(1-\mu^2)\frac{\partial T(\mu,t)}{\partial \mu}\right) + A + BT = Qa(\mu,t)s(\mu,t), \quad (10)$$

where $C(\mu,t)$ is heat capacity (having different values over land, ice, and ocean). By expanding dependent variables as

$$T(\mu,t) = T_0(\mu) + T_1(\mu)e^{i\omega_1 t} + T_2(\mu)e^{i\omega_2 t} + \cdots, \quad (11a)$$

$$s(\mu,t) = s_0(\mu) + s_1(\mu)e^{i\omega_1 t} + s_2(\mu)e^{i\omega_2 t} + \cdots, \quad (11b)$$

where $\omega_1$ and $\omega_2$ are the frequencies of sinusoidal components. Inserting (11) into (10) (with the assumption that the model parameters are independent of time) and taking the zeroth component (mean) of the resulting equation, we have

$$-\frac{\partial}{\partial \mu}\left(D(1-\mu^2)\frac{\partial T_0(\mu)}{\partial \mu}\right) + A + BT_0 = Qa(\mu)s_0(\mu). \quad (12)$$

Equation (12) is equivalent to (6) as far as the annual mean temperature is concerned. Obviously, an adequate explanation is needed in terms of how Figure 3 is produced.

**7**. Figure 3: The main effect of heat capacity in the original EBM is in the context of the amplitude of the annual and semi-annual cycles (see North and Kim,

p152).  The annual and semi-annual cycles are seriously affected by the choice of heat capacity, whereas the annual mean component is not.  The author should demonstrate that not only annual-mean temperature distribution but also the annual and semi-annual cycles of temperature is reproduced reasonably by their choice of heat capacity (see, for example, Fig. 6.8 of North and Kim).

**8**.  P8 L20:  What does the first law of thermodynamics have anything to do with incoming radiation = outgoing radiation?  Is the author referring to the zeroth law of thermodynamics?

**9**.  P8 L22-24:  As already demonstrated in Comment #2, global average temperature is close to the observed value without the effect of heat capacity.  Further, by using diffusive heat transport, zonally average temperature is reproduced close to actual observation in Comment #3.

**10**.  It is difficult to review the entire manuscript until my earlier comments are fully addressed.  In particular, the author needs to explain clearly how the solutions in each figure are computed (with appropriate equation if possible) and demonstrate clearly that his full solution (with diurnal and annual cycles) matches reasonably with the observations for his choice of heat capacity.

**Technical Comments**

1.  There is, in general, lack of explanation for variables used in the equations.

2.  P4 L6?:  "shown in Fig. 2 as the  red line with the mean …"

---

## Author Comment (AC2) · 24 Nov 2019

Thanks for the detailed critics on the *Interactive comment on Earth Syst. Dynam. Discuss., https://doi.org/10.5194/esd-2019-35, 2019.* by referee #2. The comments will be helpful in improving the readability of the manuscript. In the following, I will repeat and answer to these comments. Furthermore, possible action is proposed.

**Comment 1**

Eq. (4)

$$\epsilon\sigma T^4(\Theta,\varphi) \quad = \quad (1-\alpha)S \quad \cos\varphi \quad \cos\Theta \quad \times \quad 1_{[-\pi/2<\Theta<\pi/2]}(\Theta)$$

is not a good description of the energy balance on the surface of the earth. A description of the spatial distribution of energy requires an introduction of energy redistribution by ocean currents, eddies, etc. Further, sun's declination angle should be taken into account in order to describe reasonable spatial distribution of energy at any specific time of the year. What is described here is, at best, an energy balance in an annual-mean sense when there is no physical mechanism for redistribution of strong energy surplus in the equatorial region and strong energy deficit in the polar region.

**Answer/Action**

The left-hand side of (4) as well as the right-hand side are latitude $\varphi$- and longitude $\Theta$-dependent. The incoming radiation goes with the cosine of latitude and longitude, and there is only sunshine during the day. This is noted as the $1_{[-\pi/2<\Theta<\pi/2]}(\Theta)$ function which is zero outside the interval $[-\pi/2 < \Theta < \pi/2]$. The global mean temperature is not affected by the obliquity and precession (Berger and Loutre 1991; 1997; Laepple and Lohmann 2009). Therefore, we ignore the Earth orbital parameters for a while which makes an analytical calculation possible. Later in the numerical treatment, the full seasonal cycle is taken into account (equation (16) in the manuscript). Indeed, eq. (4) can be better explained and motivated.

**Comment 2**

Eq. (8): This is a strange derivation. Let us consider outgoing longwave radiation and incoming solar radiation in the form

$$\epsilon\sigma T^4(\phi,\Theta) \quad = \quad (1-\alpha)S \quad \cos\phi \quad \cos\Theta \quad \times \quad 1_{[-\pi/2<\Theta<\pi/2]}(\phi)$$

where $\phi$ is longitute and $\Theta$ is latitude. Equation (1) defines energy per unit time per unit area as the dimension of $\sigma = 5.670373 \times 10^{-8} Wm^{-2}K^{-4}$ indicates. Thus, total

incoming radiation can be written as . . .

$$(1 - \alpha)S \ \pi \ R^2$$

Similarly, total outgoing radiation can be written as . . .

$$\epsilon\sigma\overline{T}^4 \ 4\pi \ R^2$$

Thus, we arrive at

$$\overline{T} = \sqrt[4]{\frac{(1 - \alpha)S}{4\epsilon\sigma}}$$

**Answer/Action**

Your equation above (in your review eq. (1)), is basically the same as the one I used. The difference is that you defined $\phi$ as longitude and $\Theta$ as latitude, whereas I do it in the other way round. (By the way: This is the same approach you criticized in your comment #1.)

The calculation you presented is also more or less the same as mine, with one **fundamental** difference. In my derivation by integration over the Earth surface

$$\epsilon\sigma 4\pi\overline{T^4} = (1 - \alpha)S \ \pi \ .$$

In this formula, the average $\overline{T^4}$ is calculated, not $\overline{T}^4$. Therefore,

$$\overline{T} = \sqrt[4]{\frac{(1 - \alpha)S}{4\epsilon\sigma}}$$

is not correct. This argument to exchange the $\overline{T^4}$ by $\overline{T}^4$ was one of the motivation to write down the global energy balance in a correct form. In order to $\overline{T}$, one has a

more lengthy calculation: If we now calculate the zonal mean of the temperature by integration at the latitudinal cycles we have

$$
\begin{aligned}
T(\varphi) &= \frac{1}{2\pi} \int_{-\pi/2}^{-\pi/2} \sqrt[4]{\frac{(1-\alpha)S\cos\varphi\cos\Theta}{\epsilon\sigma}} d\Theta \\
&= \frac{\sqrt{2}}{2\pi} \underbrace{\int_{-\pi/2}^{\pi/2} (\cos\Theta)^{1/4} d\Theta}_{=\sqrt{\pi}\Gamma(5/8)/\Gamma(9/8)} \sqrt[4]{\frac{(1-\alpha)S}{4\epsilon\sigma}} (\cos\varphi)^{1/4} \\
&= \frac{1}{\sqrt{2\pi}} \frac{\Gamma(5/8)}{\Gamma(9/8)} \cdot \sqrt[4]{\frac{(1-\alpha)S}{4\epsilon\sigma}} (\cos\varphi)^{1/4} \\
&= 0.608 \cdot \sqrt[4]{\frac{(1-\alpha)S}{4\epsilon\sigma}} (\cos\varphi)^{1/4}
\end{aligned}
$$

as a function of latitude. $\Gamma$ is Euler's Gamma function with $\Gamma(x+1) = x\Gamma(x)$. When we integrate this over the latitudes, we obtain

$$
\begin{aligned}
\overline{T} &= \frac{1}{2} \int_{-\pi/2}^{\pi/2} T(\varphi)\cos\varphi\, d\varphi \\
&= \frac{1}{2} \frac{\Gamma(5/8)}{\sqrt{2\pi}\,\Gamma(9/8)} \cdot \sqrt[4]{\frac{(1-\alpha)S}{4\epsilon\sigma}} \underbrace{\int_{-\pi/2}^{\pi/2} (\cos\varphi)^{5/4} d\varphi}_{=\sqrt{\pi}\Gamma(9/8)/\Gamma(13/8)}
\end{aligned}
$$

$$= \frac{1}{2} \frac{1}{\sqrt{2}} \frac{\Gamma(5/8)}{\Gamma(13/8)} \sqrt[4]{\frac{(1-\alpha)S}{4\epsilon\sigma}} = \frac{\sqrt{2}}{4} \frac{8}{5} \sqrt[4]{\frac{(1-\alpha)S}{4\epsilon\sigma}}$$

$$= 0.4\sqrt{2} \sqrt[4]{\frac{(1-\alpha)S}{4\epsilon\sigma}} = 0.566 \sqrt[4]{\frac{(1-\alpha)S}{4\epsilon\sigma}}$$

As a side remark: The fact that $\sqrt[4]{\overline{T^4}}$ is higher than $\overline{T}$ is consistent with Hölder's inequality (Rodgers, 1888; Hölder 1889; Hardy et al., 1934, Kuptsov, 2001).

**Comment 3**

Specific heat is not needed to reproduce the reasonable spatial distribution of temperature. For example, 1D energy balance model with meridional heat flux can be written as (see North and Kim, 2017; p123-134)

$$-\frac{d}{d\mu} \left( D(1-\mu^2)\frac{dT(\mu)}{d\mu} \right) + A + BT(\mu, t) = Qa(\mu)s(\mu)$$

. . .

with a realistic insolation distribution function and a realistic albedo (see North and Kim, 2017 for details), we obtain a solution as in Figure 1. The model solution is fairly similar to the observational data. Further, the diffusive heat transport in a zonal mean sense looks very reasonable compared to that derived from satellite observations (see Figure 2).

**Answer/Action**

Thanks for your comment and hinting to the important work of North and Kim (2017). As I mention in the manuscript: "The linearization of the long wave radiation in several models (North et al., 1975a, b; Chen et al., 1995) implicitly assumes the above

heat capacity and fast rotation arguments. " Indeed, the linearized version can give a reasonable zonal mean climate.

In the revised version, I can stress out this more clearly. My point is that we need a rapidly rotating object with significant heat capacity. Without these effects, the global mean temperature would be lower. I will explicitly show that these effects are important for the radiation balance, other processes - like horizontal transport processes - are only of secondary importance for the globally averaged temperature. The finding is furthermore important to see that the effective heat capacity (which is time-scale dependent) has a direct influence on the global (and regional) temperatures). In the revised version, I will include a figure to show how the temperature would change in a slowly rotating planet.

**Comment 4**

"The atmospheric circulation provides an efficient way to propagate heat along latitudes which is ignored and is a second order effect (not shown)." This statement is erroneous. As demonstrated in Comment #3 above, a reasonable temperature distribution on the surface of the earth is reproduced by using diffusive heat transport. Heat capacity is not even used in this calculation of equilibrium temperature.

**Answer/Action**

Yes, this can be better motivated. Indeed, the energy input is time-dependent. Therefore, the mean and latitudinal temperature depends on the time derivative. See also my comments and action points in answer to Comment #3. As seen in the manuscript, the temperatures do depend on the effective heat capacity.

**Comment 5**
Eq. (12): The author introduced diurnal cycle of temperature and determined the global average of the averaged diurnal cycle of temperature. This discussion is erroneous. We can write diurnal temperature change as . . .

Further, incoming solar radiation has the same form as in (1). Thus, we arrive at the same conclusion as in Comment #2.

**Answer/Action**

Thanks for making this hint. Indeed, I agree to your basic equation (1), but not with the conclusion. See my answer to your Comment #2.

**Comment 6**

A time-dependent 1D EBM can be written as . . .

Equation (12) is equivalent to (6) as far as the annual mean temperature is concerned. Obviously, an adequate explanation is needed in terms of how Figure 3 is produced.

**Answer/Action**

I agree for the linearized EBM. See my answer to your Comment #3. I will explicitly state that in the linearized EBM, the heat capacity argument is implicitly included. Furthermore, parameter study of Figure 3 will be better explained. The model that I use is the non-linear model with the $T^4$-term and a time-dependent forcing.

**Comment 7**

Figure 3: The main effect of heat capacity in the original EBM is in the context of the amplitude of the annual and semi-annual cycles (see North and Kim, p152). The annual and semi-annual cycles are seriously affected by the choice of heat capacity,

whereas the annual mean component is not. The author should demonstrate that not only annual-mean temperature distribution but also the annual and semi-annual cycles of temperature is reproduced reasonably by their choice of heat capacity (see, for example, Fig. 6.8 of North and Kim).

**Answer/Action**

Thanks again for your comment and for hinting at the important work of North and Kim (2017). Yes, I agree for the linearized EBM. See my answer to your Comment #3 and #6. In the non-linear model, the temperature is affected by the heat capacity. The effective heat capacity is important as it is reflective of the rate of penetration of heat energy into the ocean in response to the particular pattern of forcing and the background state (Schwartz, 2007). I will also emphasize the relevance of this for climate warming scenarios. In the revised version, I will explicitly show the results of the linearized EBM.

**Comment 8**

P8 L20: What does the first law of thermodynamics have anything to do with incoming radiation = outgoing radiation? Is the author referring to the zeroth law of thermodynamics?

**Answer/Action**

Thanks. This is indeed not well formulated. In the revised version, I will explicitly state this point.

**Comment 9**

As already demonstrated in Comment #2, global average temperature is close to the

observed value without the effect of heat capacity. Further, by using diffusive heat transport, zonally average temperature is reproduced close to actual observation in Comment #3.

**Answer/Action**

Thanks. I disagree and point to my answers to Comments #2 and #3.

**Comment 10**

It is difficult to review the entire manuscript until my earlier comments are fully addressed. In particular, the author needs to explain clearly how the solutions in each figure are computed (with appropriate equation if possible) and demonstrate clearly that his full solution (with diurnal and annual cycles) matches reasonably with the observations for his choice of heat capacity.

**Answer/Action**

I agree that it seems a misunderstanding. My answers to Comments #2 and #3 shall clarify the mistake in the calculation of the temperature. The time dependence will be more explicitly stated. Furthermore, the non-linearity in the outgoing radiation makes this model different from your EBMs. The equations for the EBMs used are in the manuscript (9, 15, 16). For the equations, it is important to explicitly spell out the assumptions made. See my answer to your Comments #3 and #6. I will explicitly state that in the linearized EBM, the heat capacity argument is implicitly included. The model that I use is the non-linear model with the $T^4$-term and a time-dependent forcing: my equations (9, 15, 16).

In (16), I wrote that we include an explicit seasonal cycle into the EBM:

$$C_p\,\partial_t T \;=\; \nabla \cdot HT \;+\; (1-\alpha)S(\varphi,t) \;-\; \epsilon\sigma T^4 \quad .$$

with $S(\varphi, t)$ being calculated daily (Berger and Loutre, 1991; 1997).

In (9), the daily cycle is resolved through the time-dependence of the $1_{[-\pi/2 < \Theta < \pi/2]}(\Theta)$-term in

$$C_p \, \partial_t T \;=\; (1 - \alpha) S \cos\varphi \cos\Theta \quad \times 1_{[-\pi/2 < \Theta < \pi/2]}(\Theta) \;-\; \epsilon\sigma T^4$$

This time-dependence will be stated more explicitly.

Furthermore, I will explicitly show the results of the linearized EBM in the revised version. Indeed, the linear model has some advantages because it directly indicated the climate sensitivity. However, for a rigorous derivation of the global and regional effects, the non-linear version is an advantage. Finally, it is necessary to see that $\overline{T^4}$ is not $\overline{T}^4$. Simplified and conceptual models can be used to study long-term climate (Hasselmann, 1976; Lemke, 1977; Timmermann and Lohmann, 2000; Lohmann, 2018). As pointed out in the manuscript, the effective heat capacity is important to understand past and potential future climate.

**Technical Comment 1**

There is, in general, lack of explanation for variables used in the equations.

**Answer/Action**

Thanks. I will go through all equations in the manuscript to avoid misunderstandings. In the revised manuscript, it will be clearer that I use the non-linear EBM (as in my equations (9, 15, 16), and not the linear (North et al., 1975a, b; Chen et al., 1995; Lohmann and Gerdes, 1998; North and Kim, 2017).

**Technical Comment 2**

"shown in Fig. 2 as the (read) red line with the mean . . ."

**Answer/Action**

Thanks, this will be corrected.

**References**

Berger, A. L.: Long-term variations of daily insolation and Quaternary climatic changes, J. Atmos. Sci., 35, 2362-2367, 1978.

Berger, A., and Loutre, M.F.: Insolation values for the climate of the last 10 million years, Quat. Sci. Rev., 10(4), 297 - 317, 1991. doi:10.1016/0277-3791(91)90033-Q.

Berger, A., and Loutre, M.F.: Intertropical latitudes and precessional and half-precessional cycles, Science, 278(5342), 1476 - 1478, 1997. doi:10.1126/science.278.5342.1476.

Hardy, G. H.; Littlewood, J. E.; Pólya, G., 1934: Inequalities. Cambridge University Press, pp. XII+314, ISBN 0-521-35880-9, JFM 60.0169.01

Hasselmann, K.: Stochastic climate models. Part I, Theory. Tellus, 6:473–485, 1976.

Hölder, O., 1889: Ueber einen Mittelwertsatz. Nachrichten von der Königl. Gesellschaft der Wissenschaften und der Georg-Augusts-Universität zu Göttingen, Band (in German), (2): 38-47, JFM 21.0260.07.

Kuptsov, L. P., 2001: Hölder inequality. In Hazewinkel, Michiel (ed.), Encyclopedia of Mathematics, Springer Science+Business Media B.V. / Kluwer Academic Publishers, ISBN 978-1-55608-010-4.

Laepple, T., and Lohmann, G.: The seasonal cycle as template for climate variability on astronomical time scales. Paleoceanography, 24, PA4201, doi:10.1029/2008PA001674, 2009.

Lemke, P.: Stochastic climate models, part 3. Application to zonally averaged energy models, Tellus, 29:5, 385-392, 1977. DOI: 10.3402/tellusa.v29i5.11371

Lohmann, G.: ESD Ideas: The stochastic climate model shows that underestimated Holocene trends and variability represent two sides of the same coin. Earth Syst. Dynam. 9, 1279-1281, 2018. doi:10.5194/esd-9-1279-2018

Lohmann, G., and Gerdes, R.: Sea ice effects on the Sensitivity of the Thermohaline Circulation in simplified atmosphere-ocean-sea ice models. J. Climate 11, 2789-2803, 1998.

North, G. R.: Analytical solution of a simple climate model with diffusive heat transport. J. Atm. Sci. 32, 1300-1307, 1975a

North, G. R.: Theory of energy-balance climate models. J. Atm. Sci. 32, 2033-2043, 1997b.

North, G. R., and Kim, K.-Y., 2017: Energy Balance Climate Models. Wiley, ISBN:9783527411320, DOI:10.1002/9783527698844

Rogers, L. J.: An extension of a certain theorem in inequalities. Messenger of Mathematics, New Series, XVII (10): 145–150, 1888. JFM 20.0254.02

Schwartz, S.E.: Heat capacity, time constant, and sensitivity of Earth's climate system. J Geophys Res 112: D24S05, doi:10.1029/2007JD008746, 2007.

Timmermann, A., and Lohmann, G.: Noise-Induced Transitions in a simplified model of the thermohaline circulation, J. Phys. Oceanogr. 30 (8), 1891-1900, 2000.
* * *

---

## Referee Comment (RC3) · Anonymous Referee #2 · 25 Nov 2019

**1**. Regarding your answer to my Comment #1, I know that I used notations different from yours. All I am trying to say is that you have to mention that the specific form insolation is for the annual average.

**2**. Regarding your answer to my Comment #2, we can show by using $T(\phi) = T_0(\cos\phi)^{1/4}$ (your equation (7)) that

$$\bar{T} = \frac{1}{2}\int_{-\pi/2}^{\pi/2} T(\phi)\cos\theta\,d\phi = \frac{T_0}{2}\int_{-\pi/2}^{\pi/2}(\cos\theta)^{5/4}\,d\phi = 0.93088\,T_0, \tag{1}$$

and

$$\overline{T^4} = \frac{T_0^4}{2}\int_{-\pi/2}^{\pi/2}\cos^2\phi\,d\phi = 0.7854\,T_0^4. \tag{2}$$

Therefore,

$$\bar{T} = 0.9309\,T_0 \quad \text{and} \quad \sqrt[4]{\overline{T^4}} = 0.9414\,T_0. \tag{3}$$

Thus, the difference between the two is only ~1%. Further, you also used a similar approximation in your derivation (see equation above (10)).

The reason why you have such a strange result is that you assumed temperature to be a step function in $\Theta$, i.e.,

$$T(\varphi,\Theta) = \sqrt[4]{\frac{(1-\alpha)S\cos\varphi\cos\Theta}{\varepsilon\sigma}} \times 1_{[-\pi/2<\Theta<\pi/2]}(\Theta). \tag{4}$$

Obviously, this is not realistic at all. You have to assume that energy absorbed during the day is not emitted immediately. Perhaps for that reason, you introduced heat capacity, which makes it possible for energy to be stored during the day and released during the night.

Now, what the author is trying to convey is a little clearer. Nonetheless, presentation of the material is still very confusing. Further, the role of heat capacity should be compared and contrasted with that of heat redistribution (not only in the meridional direction but also in the zonal direction), since the latter is already proven to produce reasonable temperature distribution on the surface of the earth.

---

## Author Comment (AC3) · 25 Nov 2019

Thanks for the quick response to my comment.

**Answer to 1**

Thanks for your comment on this. I see that I shall describe the energy balance

$$\epsilon\sigma T^4(\Theta,\varphi) \quad = \quad (1-\alpha)S \quad \cos\varphi \quad \cos\Theta \quad \times \quad 1_{[-\pi/2<\Theta<\pi/2]}(\Theta)$$

in more detail. $\cos\Theta \quad \times \quad 1_{[-\pi/2<\Theta<\pi/2]}(\Theta)$ reflects the longitude dependent part which is rotating on the daily time scale. For clarification, I shall reformulate this.

The fact that I can ignore for a moment the Sun's declination angle is that for the global

temperature, it does not matter (Berger and Loutre 1991; 1997; Laepple and Lohmann 2009). Later in the paper, the seasonal cycle is explicitly used.

**Answer to 2**

Indeed, my argument goes that $\overline{T^4}$ is not the same as $\overline{T}^4$. This is essential. When using the EBM, the global temperature is calculated to be

$$\overline{T} = 0.4\sqrt{2} \sqrt[4]{\frac{(1-\alpha)S}{4\epsilon\sigma}} = 0.566 \sqrt[4]{\frac{(1-\alpha)S}{4\epsilon\sigma}}$$

This is the result without the heat capacity. When I use the heat capacity and a weak day-to-night cycle, then (my equation (12) in the manuscript) I obtain

$$\overline{\overline{T}} = 0.989 \sqrt[4]{\frac{(1-\alpha)S}{4\epsilon\sigma}}$$

The numerical solution confirms this and is shown in Fig. 3. As seen in the figure, the solution depends heavily on the heat capacity.

In your comment to my comment, you provide a calculation that you describe with the words "we can show by using" that you probably use a different model, namely the linear EBM. Then, you can obtain a similar result than mine. This is exactly what I wrote "The linearization of the long wave radiation in several models (North et al., 1975a, b; Chen et al., 1995) implicitly assumes the above heat capacity and fast rotation arguments. " Indeed, the linearized version can give a reasonable zonal mean climate.

In your new comment, you said that "Further, the role of heat capacity should be compared and contrasted with that of heat redistribution (not only in the meridional direction but also in the zonal direction), since the latter is already proven to produce reasonable temperature distribution on the surface of the earth."

As an answer, the heat transport has no influence on the global mean temperature (as in my equation (15) and following text), but of course to the local one. The order of arguments is important:
1) heat capacity and fast rotating Earth (implicitly included in the linearized version)
2) heat transport, seasonal cycle, feedbacks (as mentioned in the manuscript)

I will clarify this in order to avoid misunderstandings.

---

## Referee Comment (RC4) · Anonymous Referee #3 · 17 Dec 2019

The manuscript presented by Gerrit Lohmann aims at investigating the role of the effective heat capcity in an EBM model (Energy Balance Model) when put in the context of the day / night succession. Starting from the classical very simplified global EBM without diurnal cycle, the author progresses into showing that when adding the the diurnal cycle, the effective heat capacity of the planet cannot be neglected. The effect obtained is however quite small on the result. In a subsequent section, the author discusses how temperature meridional gradients may be affected by the oceanic vertical diffusivity and therefore the corresponding "effective" heat capacity. Though I am in principle a big supporter of simplified modelling approaches, **I do not recommend publication of the manuscript in its present form**. Indeed, the first sections of the manuscript concentrate on very obvious results that are ill-presented while the second

part do not provide enough supporting evidence on the claims laid.

**Main comments**

1. Neglecting the diurnal cycle in EBMs is a rather standard procedure. This assumes that the Earth receives a mean daily incoming solar energy equally distributed over each latitude bands. This is indeed most of the time quite a reasonable hypothesis for such simplified models, since the ocean surface temperature diurnal changes are small (at most a few degrees). This paper confirms this usual assumption, with the red and dotted brownish curves of Figure 2 being almost indistinguishable.

   The presentation on this section is however extremely confusing. The author starts with the classical 0-dimensional time average EBM. He then presents the 1-dimensional case with a daily cycle as an extension, just introducing it as a local extension of the 0-dimensional case. However, considering that there is a local energy balance is not a valid assumption in general, contrary to the one at the global scale. Obviously, it is clearly entirely irrelevant to consider that there can be an instantaneous radiative equilibrium, with temperatures dropping to zero Kelvin as soon as the Sun sets. This is clearly not what people usually assume when using EBMs !

   The usual starting point corresponds to equations (11-12) where the solar forcing is averaged over one Earth rotation. This is more or less what people have been using in geographically explicit EBMs, including the very first ones. Budyko and Sellers 1969 where indeed geographically explicit, without a diurnal cycle, as in equation (11). The authors comes back to it as a compensation of the incoherent assumption of a local radiative equilibrium. So part 2 is just showing that an irrelevant hypothesis produces irrelevant results. It brings nothing interesting, but only confusion.
To add to the confusion some assumptions are clearly not explained. On the top of page 4, the first equation is clearly invalid unless strong hypotheses are imposed, which are not specified in the text.

2. The second part of the paper discusses the role of heat capacity in the "diurnal averaging" of temperatures. Results are summarized on Fig.3. As discussed above, the fact that temperatures are much lower for small heat capacities is rather obvious (with Earth losing most of, or all its thermal energy during the night).

3. Using the typical oceanic vertical diffusivities for estimating a heat capacity is not very relevant. The diurnal cycle is buffered by the very top layers of the ocean that are usually almost well-mixed by winds and also by the diurnal cycle itself. The interior ocean vertical diffusivity has no role.

4. I do not see what is the purpose of solving equation (15) and showing Figure 4. This does not relate to the diurnal cycle, nor to heat capacity, nor to vertical mixing. What is the point ? The statement "global mean temperature is not affected by the transport because of the boundary condition...." is a bit strange. I would write more simply that here, global mean temperature is a measure of global heat content (uniform heat capacity) which depends only of global net radiative fluxes, not internal redistribution.

5. Bottom of page 5 *"Until now we assumed that the Earth's axis ..."*: It is quite awkward to explain only now where equation (4) page 2 comes from. Indeed equation (4) is certainly not standard for the Earth in particular in the context of EBMs and climate modeling. A planet with no tilt has no seasonal cycle. Many

Interactive
comment

EBMs have an explicit seasonal cycle. Again, the starting point of the paper is very awkward.

6. In the last part, the author presents some experiments with the COSMOS coupled model, to investigate the role of vertical mixing on the meridional temperature gradient. Unfortunately, it is not clear at all that these results are linked to the diurnal cycle or heat capacity. The author sets an experiment with an 25-fold increase in the background diffusivity. The logical outcome of this experiment should be to increase dramatically the oceanic circulation (not shown in the manuscript) and thus to increase massively the heat transport and the vertical mixing in the ocean. How does this relate to the heat capacity or the diurnal cycle is a mystery for me and how conclusions can be drawn from there is likewise impossible to understand. The only clear result is a weakening of the equator-to-pole gradient (likely due to increase heat transport by the ocean); however there is no physical basis to link this to past climates as the author is doing, since no probable mechanism can be suggested to increase the diffusivity by a factor 25 globally.

7. In the conclusion, there are some mentions of possible linearization of the long wave radiation. Since this is critical to the whole paper (averaging T is not the same as averaging $T^4$), I am surprised not to see a much more detailed discussion of this point much earlier in the paper.

---

## Author Comment (AC5) · 5 Jan 2020

Thanks for the constructive critics in the reviews. I realized two major common points that are addressed here.

1) The first is on the theoretical arguments, in particular, equation (4) in the paper. In the enclosed figure, I show the lat-lon dependence of the incoming short wave radiation. We assume an idealized geometry of the Earth. The global mean temperature is not affected by the obliquity and precession (Berger and Loutre 1991; 1997; Laepple and Lohmann 2009). The beauty of equation (4) is that we ignore the Earth orbital

parameters first (no obliquity and no precession) which makes an analytical calculation possible. The incoming radiation goes with the cosine of latitude and longitude, and there is only sunshine during the day.

To illustrate equation (4), I refer to Fig. R1. Panel a shows the latitudinal dependence. As we assume no tilt (this assumption is later relaxed), the latitudinal dependence is a function of latitude only: $\cos\varphi$. On the right-hand side, the function is shown. Panel b shows the latitudinal dependence is a function of longitude: $\cos\Theta$ for the sun-shining side of the Earth, and for the dark side of the Earth it is zero. For simplicity, we can define the angle $\Theta$ anti-clockwise on the for the sun-shining side between $-\pi/2$ and $\pi/2$. We define the maximal insolation always at $\Theta = 0$ which is moving in time. In the panel, the Earth's rotation is schematically sketched as the red arrow, and we see the time-dependence in the right-hand side. It is noted that the geographical longitude can be calculated by $mod(\Theta - 2\pi \cdot t/24, 2\pi)$ where t is measured in hours and mod is the modulo operation.

The motivation is that we may think of a climate system having a higher net heat capacity producing flat temperature gradients. The main point is, that the effective heat capacity (and its temporal variation over the daily/seasonal cycle) needs to be taken into account when estimating surface temperature from the energy budget.

Fig. 3 in the paper indicates that the temperature gradient is getting flatter for large heat capacities when analyzing the diurnal cycle. Fig. R2 shows explicitly the temperature dependence on heat capacity (and rotation rate). Later in the paper, an explicit seasonal cycle is included as

$$C_p\,\partial_t T \;=\; \nabla \cdot HT \;+\; (1-\alpha)S(\varphi,t) - \;\epsilon\sigma T^4 \quad .$$

with $S(\varphi,t)$ being calculated daily (Berger and Loutre, 1991; 1997). Figs. 5 and 6 in the paper indicate that the temperature gradient is getting flatter for large heat capacities associated with a reduced seasonal cycle. A similar feature is found in the circulation model (see my comments below).

2) The second comment is related to the statement that the linearization of the long-wave radiation in several energy balance models "implicitly assumes the above heat capacity and fast rotation arguments". Indeed the linearization of $\epsilon\sigma T^4$ is performed around $0°C$ (North et al., 1975a, b; Chen et al., 1995; Lohmann and Gerdes, 1998; North and Kim, 2017) and is formulated as $A + B \cdot T'$ with $T'$ being measured in $°C$.

As the temperatures based on the local energy balance without a heat capacity would vary between $T_{min} = 0$ K and $T_{max} = \sqrt[4]{\frac{(1-\alpha)S}{\epsilon\sigma}} = \sqrt{2} \cdot \sqrt[4]{\frac{(1-\alpha)S}{4\epsilon\sigma}} = \sqrt{2} \cdot 288K = 407$ K, a linearization would be not permitted. If the assumptions of fast rotation rate and heat capacity are ignored, the global mean temperature would be much lower as calculated in equation (8) of the paper and shown in the figures (Figs. 2 and 3, Fig. R2).

There is also a second point of why linearization has to be used carefully. As seen in Figs. 5 and 6 in the paper, the temperature gradient is getting flatter for large heat capacities. A reduced seasonal cycle is responsible for significant warming. The larger the seasonal contrast, the colder is the climate. Let us define $\overline{\cdot}$ as the averaging over a time period (in our case the seasonal or diurnal cycle), then $\overline{T^4} > \overline{T}^4$ which is consistent with Hölder's inequality (Rodgers, 1888; Hölder 1889; Hardy et al., 1934, Kuptsov, 2001). This feature is missing in the linearized version with $A + B \cdot T'$.

The climate model experiment with enhanced mixing is admittedly highly idealized. Due to higher mixing, more heat can be taken by the climate system, the seasonal cycle is reduced (Fig. 7 in the paper), and especially the high latitudes warm-up (Fig. R3). (In this circulation model, the heat takeup goes to the deeper levels in the ocean but it is not finalized even after 1000 years (Fig. R4). ) The pronounced winter warming provides an important non-linearity (cf. Hölder's inequality): Let us denote $T_{summer}$ and $T_{winter}$ for the local summer and winter temperatures, then $0.5 \cdot (T_{summer}^4 + T_{winter}^4)$ is much greater and therefore associated to a colder climate than a climate under a reduced seasonal cycle with the extreme case $(0.5 \cdot T_{summer} + 0.5 \cdot T_{winter})^4$.

**References**

Berger, A. L.: Long-term variations of daily insolation and Quaternary climatic changes, J. Atmos. Sci., 35, 2362-2367, 1978.

Berger, A., and Loutre, M.F.: Insolation values for the climate of the last 10 million years, Quat. Sci. Rev., 10(4), 297 - 317, 1991. doi:10.1016/0277-3791(91)90033-Q.

Berger, A., and Loutre, M.F.: Intertropical latitudes and precessional and half-precessional cycles, Science, 278(5342), 1476 - 1478, 1997. doi:10.1126/science.278.5342.1476.

Hardy, G. H.; Littlewood, J. E.; Pólya, G., 1934: Inequalities. Cambridge University Press, pp. XII+314, ISBN 0-521-35880-9, JFM 60.0169.01

Hölder, O., 1889: Ueber einen Mittelwertsatz. Nachrichten von der Königl. Gesellschaft der Wissenschaften und der Georg-Augusts-Universität zu Göttingen, Band (in German), (2): 38-47, JFM 21.0260.07.

Kuptsov, L. P., 2001: Hölder inequality. In Hazewinkel, Michiel (ed.), Encyclopedia of Mathematics, Springer Science+Business Media B.V. / Kluwer Academic Publishers, ISBN 978-1-55608-010-4.

Laepple, T., and Lohmann, G.: The seasonal cycle as template for climate variability on astronomical time scales. Paleoceanography, 24, PA4201, doi:10.1029/2008PA001674, 2009.

Lohmann, G., and Gerdes, R.: Sea ice effects on the Sensitivity of the Thermohaline Circulation in simplified atmosphere-ocean-sea ice models. J. Climate 11, 2789-2803, 1998.

North, G. R.: Analytical solution of a simple climate model with diffusive heat transport. J. Atm. Sci. 32, 1300-1307, 1975a

North, G. R.: Theory of energy-balance climate models. J. Atm. Sci. 32, 2033-2043, 1997b.

North, G. R., and Kim, K.-Y., 2017: Energy Balance Climate Models. Wiley, ISBN:9783527411320, DOI:10.1002/9783527698844

Rogers, L. J.: An extension of a certain theorem in inequalities. Messenger of Mathematics, New Series, XVII (10): 145–150, 1888. JFM 20.0254.02

[Figure]

**Fig. 1.** Fig. R1. Latitudinal (a) and longitudinal (b) dependence of the incoming short wave radiation. See text for the details.

[Figure]

**Fig. 2.** Fig. R2. Temperature dependence on heat capacity (and rotation rate) when analyzing the diurnal cycle.

[Figure]

**Fig. 3.** Fig. R3. Temperature anomaly (annual mean, mean over the last 100 years of integration) using the Earth system model (see text for more details).

[Figure]

**Fig. 4.** Fig. R4. Hovmoeller-diagram of global mean potential ocean temperatures (enhanced vertical mixing in the ocean) minus control experiment (standard PI) over the time period from 1 to year 1000.

---

## Author Response (AR1)

[revised manuscript text omitted]
 &= (\underset{\sim}{1-}1-\alpha)S\underline{\cos\varphi\cos\Theta}\cdot\cos\varphi\cdot\cos\Theta \quad \underline{-\epsilon\sigma T^4} \quad \times 1_{[-\pi/2<\Theta<\pi/2]}(\Theta) \qquad \text{for } -\pi/2<\Theta<\pi/2 \\
&\equiv \qquad\qquad\qquad\qquad\qquad\qquad -\epsilon\sigma T^4 \qquad \underline{\text{elesewhere}} \qquad\qquad (9)
\end{aligned}
$$

with $C_p$ representing the heat capacity multiplied with the depth of the atmosphere-ocean layer ($C_p$ is in the order of $10^7 - 10^8\,JK^{-1}m^{-2}$). If we consider the zonal mean and averaged over the diurnal cycle, we can assume that the heat capacity is
mainly given by the  atmosphere and the uppermost ocean and soil. Observational evidence is that the diurnal variation
of the ocean surface is in the order of 0.5-3 K with highest values at favorable conditions of high insolation and low winds
(Stommel, 1969; Anderson et al., 1996; Kawai and Kawamura, 2002; Stuart-Menteth, et al. 2003; Ward, 2006).

 To simplify (9), the energy balance is integrated over the longitude and  day, and assume that the variation due to the diurnal cycle is weak. With $\tilde{T} = \frac{1}{2\pi} \int_0^{2\pi} T\, d\Theta$, we find

$$\tilde{T}(\tilde{t}) = \frac{1}{2\pi} \int_0^{2\pi} T(t)\, d\Theta \quad \text{with} \quad \tilde{T}^4 \approx \frac{1}{2\pi} \int_0^{2\pi} T^4\, d\Theta$$

$$C_p\, \partial_t\, \tilde{T}\, = (1-\alpha)S\cos\varphi \cdot \underbrace{\frac{1}{2\pi} \int_{-\pi/2}^{\pi/2} \cos\Theta\, d\Theta}_{2} - \epsilon\sigma \underbrace{\underline{T^4}\, \frac{1}{2\pi} \int_0^{2\pi} T^4\, d\Theta}_{\approx \tilde{T}^4} = (1-\alpha)\frac{S}{\pi}\cos\varphi - \epsilon\sigma\tilde{T}^4 \tag{10}$$

giving the equilibrium solution

$$\tilde{T}(\varphi) = \sqrt[4]{\frac{4}{\pi}} \cdot \sqrt[4]{\frac{(1-\alpha)S}{4\epsilon\sigma}} \ (\cos\varphi)^{1/4} \tag{11}$$

10 shown in Fig. 3 as the read line . The global mean temperature is

$$\overline{\overline{T}} = \sqrt[4]{\frac{4}{\pi}} \cdot \sqrt[4]{\frac{(1-\alpha)S}{4\epsilon\sigma}} \ \underbrace{\frac{1}{2} \int_{-\pi/2}^{\pi/2} (\cos\varphi)^{5/4} d\varphi}_{1.862\sqrt{\pi}\Gamma(9/8)/\Gamma(13/8)} = \sqrt[4]{\frac{4}{\pi}\frac{1.862}{2}} \underbrace{\sqrt{\frac{\pi}{2}\frac{\Gamma(9/8)}{\Gamma(13/8)}}}_{\approx 0.989} \cdot \sqrt[4]{\frac{(1-\alpha)S}{4\epsilon\sigma}} = 0.989\sqrt[4]{\frac{(1-\alpha)S}{4\epsilon\sigma}} \approx 28$$

 which is very similar to 288 K from (3).

A numerical solution of (9) is shown as the brownish dashed line in Fig. 3 where the diurnal cycle has been taken into account and $C_p = C_p^a$ has been chosen as the atmospheric heat capacity

$$C_p^a = c_p p_s/g = 1004\, JK^{-1}kg^{-1} \cdot 10^5 Pa/(9.81 ms^{-2}) = 1.02 \cdot 10^7 JK^{-1}m^{-2}$$

which is the specific heat at constant pressure $c_p$ times the total mass $p_s/g$. $p_s$ is the surface pressure and $g$ the gravity. The global mean temperature $\overline{T}$ is 286 K, again close to 288 K.

15 Quite often the linearization the long wave radiation $\epsilon\sigma T^4$ is linearized in energy balance models. Indeed the linearization is performed around $0°C$ (North et al., 1975a, b; Chen et al., 1995; Lohmann and Gerdes, 1998; North and Kim, 2017) and is formulated as $A + B \cdot T'$ with $T'$ being measured in $°C$. As the temperatures based on the local energy balance without a heat capacity would vary between $T_{min} = 0$ K and $T_{max} = \sqrt{2} \cdot 288K = 407$ K, a linearization would be not permitted. Therefore, the linearization implicitely assumes the above heat capacity and fast rotation arguments.

20 The effect of heat capacity is systematically analyzed in Fig. 4. The temperatures are relative insensitive for a wide range of $C_p$. We find a severe drop in temperatures for heat capacities below 0.01 of the atmospheric heat capacity $C_p^a$.

furthermore  Fig. 5 shows the temperature dependence for different values of $C_p$ and the length of the day, indicating a pronounced temperature drop during night for low values of heat capacities and for  (hypothetical) long days of 240 h instead of 24 h  . We have chosen this feature for a particular latitude (here: $45^0$N). The analysis shows that the effective heat capacity is of great importance for the temperature, this depends on the atmospheric planetary boundary layer (how well-mixed with small gradients in the vertical) and the depth of the mixed layer in the ocean

~~where $k_v = k^o/C_p^o$ is the oceanic vertical eddy diffusivity in $m^2 s^{-1}$, and $C_p^o$ the oceanic heat capacity relevant on the specific time scale. The vertical eddy diffusivity $k_v$ can be estimated from climatological hydrographic data (Olbers et al., 1985; Munk and Wunsch, 1998) and varies roughly between $10^{-5}$ and $10^{-4} m^2 s^{-1}$ depending on depth and region. A scale analysis of (15) yields a characteristic depth scale $h_T$ through For the diurnal cycle $h_T$ is less that half a meter and the heat capacity generally less than that of the atmosphere. As pointed out by Schwartz (2007), the effective heat capacity that reflects only that portion of the global heat capacity that is coupled to the perturbation on the timescale of the perturbation. We discuss the sensitivity of the system with respect to $k_v$ later~~in the context of a full circulation model which will be analyzed later.

**4 Meridional temperature gradients**

Equation (10) shall be the starting point for further investigations. One can easily include the  meridional heat transport by diffusion which has been previously used in one-dimensional EBMs (e.g. Adem, 1965; Sellers, 1969; Budyko, 1969; North, 1975a,b). In the following we will drop the tilde sign. Using a diffusion coefficient k, the meridional heat transport across a latitude is $HT = -k\nabla T$. One can solve the EBM

$$C_p \partial_t T = \nabla \cdot HT + (1-\alpha)\frac{S}{\pi}\cos\varphi - \epsilon\sigma T^4 \quad . \tag{13}$$

numerically. The boundary condition is that the HT at the poles vanish. The values of k are in the range of earlier studies (North, 1975a,b; Stocker et al., 1992; Chen et al., 1995; Lohmann et al., 1996). Fig. 6 shows the equilibrium solutions of (13) using different values of $k$ (solid lines). The global mean temperature is not affected by the transport term because of the boundary condition with zero heat transport at the poles. The same is true if we introduce zonal transports because of the cyclic boundary condition in $\theta-$direction.

Until now, we assumed that the Earth's axis of rotation were vertical with respect to the path of its orbit around the Sun. Instead Earth's axis is tilted off vertical by about 23.5 degrees. As the Earth orbits the Sun, the tilt causes one hemisphere to receive more direct sunlight and to have longer days. This is a redistribution of heat with more solar insolation at the poles and less at the equator (formally it could be associated to an enhanced meridional heat transport HT). The resulting temperature is shown as the dotted blue line in Fig. 6. A spatially constant temperature in (1) can be formally seen as a system with infinite diffusion coefficient $k \to \infty$ (black line in Fig. 6).

The global mean temperatures are not affected by the tilt and the values are identical to the one calculated in (12). This is true even if we calculate the seasonal cycle (Berger and Loutre, 1991; 1997; Laepple and Lohmann, 2009). However, if we include non-linearities such as the ice-albedo feedback ($\alpha$ as a function of T), the global mean value is changing (Budyko, 1969; Sellers, 1969; North et al., 1975a, b), cf. the dashed blue line in Fig. 6. Such model can be improved by including an explicit spatial pattern with a seasonal cycle to study the long-term effects of climate to external forcing (Adem, 1981; North et al., 1983) or by adding noise mimicking the effect of short-term features on the long-term climate (Hasselmann, 1976; Lemke, 1977; Lohmann, 2018).

As a logical next step, let us now include an explicit seasonal cycle into the EBM:

$$C_p \partial_t T = \nabla \cdot HT + (1 - \alpha)S(\varphi, t) - \epsilon\sigma T^4 \quad . \tag{14}$$

with $S(\varphi, t)$ being calculated daily (Berger and Loutre, 1991; 1997). Eq. (14) is calculated numerically for fixed diffusion coefficient $k = 1.5 \cdot 10^6 m^2/s$ under present orbital conditions. Fig. 7 indicates that the temperature gradient is getting flatter for large heat capacities. Furthermore, the mean temperature is affected by the choice of $C_p$. In the case of large heat capacity at high latitudes (for latitudes polewards of $\varphi = 50°$) and moderate elsewhere, we observe strong warming at high latitudes with moderate warming at low latitudes (dashed curve). This again indicates that we cannot neglect the time-dependent left hand side in the energy balance equations, both for the diurnal (9) as well as the seasonal (14) cycle for the temperature budget. In both considered cases, at strong diurnal or seasonal amplitude lowers the annual mean temperature.

Fig. 8 shows the seasonal amplitude for the $C_p$-scenarios as indicated by the blue and dashed black lines, respectively. ~~A change in the seasonal /diurnal cycle of $T_1 - T_2 = 50°C$ is equivalent to about $10 Wm^{-2}$ when applying the long wave radiation change $\epsilon\sigma \cdot 0.5(T_1^4 + T_2^4) - \epsilon\sigma \cdot (0.5 \cdot (T_1 + T_2))^4$ for typical temperatures on the Earth. Please note that the number $10 Wm^{-2}$ is equivalent to a greenhouse gas forcing of more than quadrupling the $CO_2$ concentration in the atmosphere.~~

**5**

 The larger the seasonal contrast, the colder is the climate. Let us define here $\bar{\cdot}$ as the averaging over a time period (here the seasonal cycle), then $\overline{T^4} > \overline{T}^4$ which is consistent with Hölder's inequality (Rodgers, 1888; Hölder 1889; Hardy et al.,  1934, Kuptsov, 2001). It is noted that this feature is missing in the linearized version $A + B \cdot T'$ of the outgoing radiation. We see the large variation in the seasonal cycle $\Delta T = T_{summer} - T_{winter}$ for the blue line in Fig. 8 as compared to the dashed line. A mean change in the net long wave radiation can be approximated by the mean of summer and winter values $\epsilon\sigma \cdot 0.5(T_{summer}^4 + T_{winter}^4)$, which is up to $10 Wm^{-2}$ higher than $\epsilon\sigma \cdot (0.5 \cdot (T_{summer} + T_{winter}))^4$ if the seasonal cycle is damped as in the dashed line of Fig. 8. This implies that a lower seasonal cycle provides for a significant warming. If we would

[revised manuscript text omitted]

5 ~~There is a range of literature on the parameterisations of vertical mixing in ocean circulation models (e.g., Mellor and Yamada, 1974; Philander and Pacanowksi, 1981; Luyten et al., 1983; Large et al., 1994) and a detailed discussion is not given here. The mixing has a background value plus a mixing process strongly influenced by the shears of the mean currents. Although observations give a range of values of $k_v$ for the ocean interior, models use simplified physics and prescribe a constant background value. In numerical modelling, the values are also constrained by the required numerical stability and to fill gaps left~~

10  Interestingly, it has been suggested that the tight link between ocean temperature and $CO_2$ formed only during the Pliocene when the thermocline shoals and surface water became more sensitive to $CO_2$ (La Riviere et al., 2012) which is therefore of major importance for the understanding of the climate-carbon cycle (Wiebe and Weaver, 1999; Zachos et al.,

15  2008; de Boer and Hogg, 2014~~; de Lavergne et al., 2016; Hutchinson et al., 2018). Lambeck (1977) calculated the total rate of energy dissipation based on the numerical tide model, indicating a dominant role of ocean continent geometry and sea floor topography. Visser (2007) discussed potential biomixing in the oceans and concluded that the mixing efficiency of small organisms is extremely low. Most of the mechanical energy they impart to the oceans isdissipated almost immediately as heat. There may be a case to be made for biomixing by larger animals on a~~

20 ~~local scale, but their relatively low abundance means that they are unlikely to be important contributors to global circulation. It has to be explored if the marine organisms could have been different in the distant past in order to introduce significant changes in mixing. As a next step, one can change the large-scale ocean gateways and changed orography/topography in order to realistically simulate past climates and to separate potential forcing factors.~~ ).

**6 Conclusions**

25

This manuscript revisits the relationship between the (global mean) surface temperature of the Earth and its radiation budget as is frequently used in Energy balance models (EBMs). The main point is, that the effective heat capacity and its temporal variation over the daily/seasonal cycle needs to be taken into account when estimating surface temperature from the energy budget

30 . EMBs provide a crucial tool in climate research, especially because they - confirmed by the results of the elaborate realistic climate models - describe the processes essential for the genesis of the global climate. EBMs are thus an admissible conceptual tools, due to its reduced complexity to the essentials "scientific understanding" represents (von Storch et al., 1999). This understanding states that the radiation balance on the ground and

the absorption in the atmosphere are the essential factors for determining the temperature. Eq. (3) says that the temperature is independent of the size of the Earth and the thermal characteristics, but depends on the albedo, emissivity and solar constant.

The argument follows the  $1^{st}$ law of thermodynamics on the conservation of energy: in steady state the Earth has to emit as much energy as it receives from the Sun. However, I argue that we shall explicitly emphasize the Earth as a rapidly
5  rotating object with a significant heat capacity in our EBMs. Without these effects, the global mean temperature would be much lower. This description can be better used for objects like the Moon or Mercury (Vasavada et al., 1999) as slowly rotating bodies without significant heat capacity. The Earth system understanding says that these effects are important for the radiation balance, other processes - like horizontal transport processes - are only of secondary importance for the globally averaged temperature. The linearization of the long wave radiation in several models (North et al., 1975a, b; Chen
10  et al., 1995) implicitly assumes the above heat capacity and fast rotation arguments.  Ironically, the global mean temperature in the revised EBM is very close to the original proposed value.
15

As a basic feature, we detect the strong dependence of the temperature distribution on the effective heat capacity linked to the mixed-layer depth. A change in the mixed layer depth which likely happened through glacial-interglacial cycles (e.g. Zhang et al., 2014)  can therefore an important driver constraining climate sensitivity (Köhler, et al., 2010). This could be also relevant for future climate change when the ocean stratification can change.
20

~~As a key aspect for climate sensitivity, La Riviere et al. (2012) have claimed that the tight link between ocean temperature and CO$_2$ formed only during the Pliocene when the thermocline shoals and surface water became more sensitive to CO$_2$ which is therefore of major importance for the understanding of the climate-carbon cycle (Wiebe and Weaver, 1999; Zachos et al., 2008; de Boer and Hogg, 2014). Schwartz (2007) stressed out that the effective heat capacity is not an intrinsic property of the~~
25  ~~climate system but is reflective of the rate of penetration of heat energy into the ocean in response to the particular pattern of forcing and -as suggested by La Riviere et al. (2012)- also to the background state. As one application, we change the vertical mixing in the ocean affecting the effective heat capacity. The resulting temperature might explain 
[revised manuscript text omitted]

Hutchinson, D. K., A. M. de Boer, H. K. Coxall, R. Caballero, J. Nilsson, and M. Baatsen: Climate sensitivity and meridional overturning circulation in the late Eocene using GFDL CM2.1. Clim. Past, 14, 789-810, 2018.

20     IPCC: Climate Change: The Physical Science Basis. Contribution of Working Group I to the Fifth Assessment Report of the Intergovernmental Panel on Climate Change Stocker, T.F., D. Qin, G.-K. Plattner, M. Tignor, S.K. Allen, J. Boschung, A. Nauels, Y. Xia, V. Bex and P.M. Midgley (eds.). Cambridge University Press, Cambridge, United Kingdom and New York, NY, USA, 1535 pp, 2013.

[revised manuscript text omitted]

Rahmstorf, S., M. Crucifix, A. Ganopolski, H. Goosse, I. Kamenkovich, R. Knutti, G. Lohmann, B. Marsh, L. A. Mysak, Z. Wang, A. Weaver, 2005: Thermohaline circulation hysteresis: a model intercomparison. Geophys. Res. Lett., 32, L23605, doi:10.1029/2005GL023655.

[revised manuscript text omitted]

35  von der Heydt, A.S., H.A. Dijkstra, R.S.W. van de Wal, R. Caballero, M. Crucifix, G.L. Foster, M. Huber, P. Köhler, E. Rohling, P. J. Valdes, P. Ashwin, S. Bathiany, T. Berends, L. G. J. van Bree, P. Ditlevsen, M. Ghil, A. Haywood, J. Katzav, G. Lohmann, J. Lohmann,

V. Lucarini, A. Marzocchi, H. Pälike, I.R. Baroni, D. Simon, A. Sluijs, L.B. Stap, A. Tantet, J. Viebahn, M. Ziegler: Lessons on climate sensitivity from past climate changes. Current Climate Change Reports 2, 148?158. DOI:10.1007/s40641-016-0049-3, 2016.

von Storch, H., Güss, G. S., and Heimann, M.: Das Klimasystem und seine Modellierung: eine Einführung. (in German) Springer-Verlag Berlin Heidelberg. 256pp. doi:10.1007/978-3-642-58528-9, 1999.

5   Visser, A.W. : Biomixing of the Oceans? Science 316, 5826, 838-839. DOI:10.1126Wang, Z., Schneider, E. K., and Burls, N. J.: The sensitivity of climatological SST to slab ocean model thickness. Climate Dynamics, 1-15, 2019. http:/science.1141272, 2007. /doi.org/10.1007/s00382-019-04892-0

Ward, B.: Near-surface ocean temperature. J Geophys Res 111(C5):1-18. doi:10.1029/2004JC002689, 2006.

Wei, W., and Lohmann, G.: Simulated Atlantic Multidecadal Oscillation during the Holocene. J. Climate, 25, 6989-7002, 2012.

10  Wiebe, E.C., and Weaver, A. J.: On the sensitivity of global warming experiments to the parameterisation of sub-grid scale ocean mixing. Climate Dyn., 15, 875-893, 1999.

Wolfe, J.A.: Tertiary climatic changes at middle latitudes of western North America. Palaeogeography, Palaeoclimatology, Palaeoecology 108, 195-205, 1994.

Wunsch, C., and Ferrari, R.: Vertical mixing, energy, and the general circulation of the oceans. Annu. Rev. Fluid Mech. 36:281-314, 2004.

Zachos, J.C., Dickens, G.R., and Zeebe, R.E.: An early Cenozoic perspective on greenhouse warming and carbon-cycle dynamics. Nature, 451, 279-283, 2008.

Zhang, X., Lohmann, G., Knorr, G., and Xu, X.: Different ocean states and transient characteristics in Last Glacial Maximum simulations and implications for deglaciation. Clim. Past, 9, 2319-2333, 2013.

20  Zhang, X., Lohmann, G., Knorr, G., and Purcell, C.: Abrupt glacial climate shifts controlled by ice sheet changes. Nature 512, 290-294, 2014.

**Answer to interactive comment of #referee 1 on Temperatures from Energy Balance Models: the effective heat capacity matters**

Gerrit Lohmann[1,2]

[1]Alfred Wegener Institute, Helmholtz Centre for Polar and Marine Research, Bremerhaven, Germany
[2] University of Bremen, Bremen, Germany

Thanks for the constructive critics in the *Interactive comment on Earth Syst. Dynam. Discuss., https://doi.org/10.5194/esd-2019-35, 2019.* by referee #1. In the following, I will repeat and answer to these comments. Furthermore, the actions are described.

**Comment**

5    This manuscript revisits the relationship between the (global mean) surface temperature of the Earth and its radiation budget as is frequently used in Energy balance models (EBMs). The main point is, that the effective heat capacity (and its temporal variation over the daily/seasonal cycle) needs to be taken into account when estimating surface temperature from the energy budget. The results of this exercise together with coupled ocean-atmosphere GCM simulations lets the author suggest a potential mechanism for the relatively low equator-to-pole temperature gradient in past warm climates that has been observed in

10   proxy data, but remains difficult to reproduce with GCMs. The paper includes a very useful discussion about general properties of the energy balance of the Earth and this certainly justifies publication in ESD. However, I have two main comments to be improved on before I can recommend publication.

**Comment 1a)**

The theoretical arguments should be much better explained. This holds in particular for sections 2 and 3. For example, after

15   or before eq. (4), it should be very explicitly explained which variables become lat-lon dependent, and which not. Otherwise eq. (4) and the analysis that follows is very hard to understand (or reproduce). In my view, if you consider the local energy balance, temperature T, emissivity and albedo $\alpha$, should be spatially dependent and therefore this should have consequences for the following integration. If they are not spatially dependent, then it should be clearly stated why not.

**Answer/Action**

20   Indeed, eq. (4) can be better explained. I have rewritten section 2, added a figure, and I think the notation is now clearer.

The incoming radiation goes with the cosine of latitude $\varphi$ and longitude $\Theta$, and there is only sunshine during the day. Fig. 1a shows the latitudinal dependence. As we assume no tilt (this assumption is later relaxed), the latitudinal dependence is a function of latitude only: $\cos\varphi$. On the right-hand side, the function is shown. Fig. 1b shows the latitudinal dependence is a function of longitude: $\cos\Theta$ for the sun-shining side of the Earth, and for the dark side of the Earth it is zero. For simplicity,

[Figure]

**Figure 1.** New Figure 2 in the paper: Latitudinal (a) and longitudinal (b) dependence of the incoming short wave radiation. On the right hand side, the insolation as a function of latitude $\varphi$ and longitude $\Theta$ with maximum insolation $(1-\alpha)S$ is shown. See text for the details.

we can define the angle $\Theta$ anti-clockwise on the for the sun-shining side between $-\pi/2$ and $\pi/2$. We define the maximal insolation always at $\Theta = 0$ which is moving in time. In the panel, the Earth's rotation is schematically sketched as the red arrow, and we see the time-dependence in the right-hand side. It is noted that the geographical longitude can be calculated by $mod(\Theta - 2\pi \cdot t/24, 2\pi)$ where t is measured in hours and mod is the modulo operation. Summarizing our geometrical considerations, we can now write the local energy balance as

$$\epsilon \sigma T^4 = (1-\alpha)S \cdot \cos\varphi \cdot \cos\Theta \qquad \text{for } -\pi/2 < \Theta < \pi/2 \tag{1}$$

and zero during night for $\Theta < -\pi/2$ or $\Theta > \pi/2$. Temperatures based on the local energy balance without a heat capacity would vary between $T_{min} = 0$ K and $T_{max} = \sqrt[4]{\frac{(1-\alpha)S}{\epsilon\sigma}} = \sqrt{2} \cdot \sqrt[4]{\frac{(1-\alpha)S}{4\epsilon\sigma}} = \sqrt{2} \cdot 288K = 407$ K.

Integration of (1) over the Earth surface is

$$\int\limits_{-\pi/2}^{\pi/2} \left( \int\limits_{0}^{2\pi} \epsilon\sigma T^4 R\cos\varphi \, d\Theta \right) R\, d\varphi \;=\; (1-\alpha)S \int\limits_{-\pi/2}^{\pi/2} R\cos^2\varphi \, d\varphi \cdot \int\limits_{-\pi/2}^{\pi/2} R\cos\Theta \, d\Theta$$

$$\epsilon\sigma R^2 \frac{4\pi}{4\pi} \int\limits_{-\pi/2}^{\pi/2} \left( \int\limits_{0}^{2\pi} T^4 \cos\varphi \, d\Theta \right) d\varphi \;=\; (1-\alpha)S R^2 \underbrace{\int\limits_{-\pi/2}^{\pi/2} \cos^2\varphi \, d\varphi}_{\frac{\pi}{2}} \cdot \underbrace{\int\limits_{-\pi/2}^{\pi/2} \cos\Theta \, d\Theta}_{2}$$

$$\epsilon\sigma 4\pi \overline{T^4} \;=\; (1-\alpha)S\,\pi \qquad\qquad (2)$$

5   giving a similar formula as (3, in the paper) with the definition for the average $\overline{T^4}$.

What we really want is the mean of the temperature $\overline{T}$. Therefore, we take the fourth root of (1):

$$T = \sqrt[4]{\frac{(1-\alpha)S\cos\varphi\cos\Theta}{\epsilon\sigma}} \qquad \text{for } -\pi/2 < \Theta < \pi/2 \qquad\qquad (3)$$

and zero elsewhere. If we calculate the zonal mean of (3) by integration at the latitudinal cycles we have

$$T(\varphi) \;=\; \frac{1}{2\pi} \int\limits_{-\pi/2}^{-\pi/2} \sqrt[4]{\frac{(1-\alpha)S\cos\varphi\cos\Theta}{\epsilon\sigma}} \, d\Theta$$

$$\;=\; \frac{\sqrt{2}}{2\pi} \underbrace{\int\limits_{-\pi/2}^{\pi/2} (\cos\Theta)^{1/4} d\Theta}_{\sqrt{\pi}\Gamma(5/8)/\Gamma(9/8)} \sqrt[4]{\frac{(1-\alpha)S}{4\epsilon\sigma}} \;\; (\cos\varphi)^{1/4} \;=\; \underbrace{\frac{1}{\sqrt{2\pi}} \frac{\Gamma(5/8)}{\Gamma(9/8)}}_{\approx 0.608} \cdot \sqrt[4]{\frac{(1-\alpha)S}{4\epsilon\sigma}} \;\; (\cos\varphi)^{1/4} \qquad (4)$$

as a function on latitude (Fig. 3 in the paper). $\Gamma$ is Euler's Gamma function with $\Gamma(x+1) = x\Gamma(x)$. When we integrate this over the latitudes, we obtain

$$\overline{T} \;=\; \frac{1}{2} \int\limits_{-\pi/2}^{\pi/2} T(\varphi)\cos\varphi \, d\varphi \;=\; \frac{1}{2} \frac{\Gamma(5/8)}{\sqrt{2\pi}\,\Gamma(9/8)} \cdot \sqrt[4]{\frac{(1-\alpha)S}{4\epsilon\sigma}} \;\; \underbrace{\int\limits_{-\pi/2}^{\pi/2} (\cos\varphi)^{5/4} d\varphi}_{\sqrt{\pi}\Gamma(9/8)/\Gamma(13/8)}$$

$$\;=\; \frac{1}{2} \frac{1}{\sqrt{2}} \frac{\Gamma(5/8)}{\Gamma(13/8)} \cdot \sqrt[4]{\frac{(1-\alpha)S}{4\epsilon\sigma}} \;=\; \underbrace{\frac{\sqrt{2}}{4} \frac{8}{5}}_{0.4\sqrt{2}\approx 0.566} \cdot \sqrt[4]{\frac{(1-\alpha)S}{4\epsilon\sigma}} \qquad\qquad (5)$$

15   Therefore, the mean temperature is a factor $0.4\sqrt{2} \approx 0.566$ lower than $288$ K as stated at (**??**) and would be $\overline{T} = 163$ K. The standard EBM in Fig. 1 (in the paper) has imprinted into our thoughts and lectures. We should therefore be careful and pinpoint the reasons for the failure. What happens here is that the heat capacity of the Earth is neglected and there is a strong non-linearity of the outgoing radiation.

**Comment 1b)**

I find it very puzzling that the heat capacity $C_p$ does not explicitly appear in eq. (11), although I clearly see how you get there. A few words of explanation would be very useful to the (less-expert) reader.

**Answer/Action**

5    Indeed, the equilibrium solution is calculated from (10) as there is no time-dependence due to the integration over longitude and day. In the revised version, I was better to avoid the notation of the $1_{[-\pi/2<\Theta<\pi/2]}(\Theta)$ function. The formulation is now

$$C_p \partial_t T \quad = (1-\alpha)S \cdot \cos\varphi \cdot \cos\Theta \quad -\epsilon\sigma T^4 \qquad \text{for } -\pi/2 < \Theta < \pi/2$$
$$= \qquad\qquad\qquad\qquad -\epsilon\sigma T^4 \qquad \text{elesewhere}$$

Now, it becomes clear that the equilibrium solution is given. I write explicitly in line 20-21 of page 4, and line 1 on page 5

10    how the averaging is done.

**Comment 1c)**

Then, after eq. (12) the reference heat capacity is chosen as the atmospheric heat capacity. Why is that? Above in the text you have said that the heat capacity is manly given by the ocean, so why do you use the atmospheric heat capacity here?

**Answer/Action**

15    Yes, the effective heat capacity is time-scale dependent. For the day and night cycle values in the order of the atmospheric heat capacity are realistic for our Earth with 24 h rotation. In the revised version, I am more explicit here and show how the temperature would change in a slowly rotating planet (new Fig. 5).

In the section 5; I have added a discussion of the effective heat capacity. For the diurnal cycle $h_T$ is less that half a meter and the heat capacity generally less than that of the atmosphere. The seasonal mixed layer depth can be several hundred meters

20    (e.g., de Boyer Montégut et al., 2004). As pointed out by Schwartz (2007), the effective heat capacity that reflects only that portion of the global heat capacity that is coupled to the perturbation on the timescale of the perturbation. In the context of global climate change induced by changes in atmospheric composition on the decade to century timescale the effective heat capacity is subject to change in heat content on such timescales. For the seasonal cycle and longer time scales, this issue is more difficult. I tried to avoid realistic scenarios, because than one would have to define geographical details. There is no demand

25    for realism here.

**Comment 1d)**

A bit more explanation and motivation should also enter the fact that in one case in Fig. 5 you use a latitudinal dependent heat capacity (in the text just after eq. (12)). How exactly? And what is the motivation for that?

**Answer/Action**

30    Yes, this can be better motivated. The high latitudes have a much higher effective heat capacity due to the deeper mixed layer at high latitudes. It is just an assumption, see above.

**Comment 1e)**

On page 6, line 18, the temperatures T1 and T2 remain unexplained!

**Answer/Action**

In the revised version, these temperatures are better explained. It is a back-on-the-envelope calculation to see the main argument. I replaced this section by

Fig. 8 shows the seasonal amplitude for the $C_p$-scenarios as indicated by the blue and dashed black lines, respectively. ~~A change in the seasonal /diurnal cycle of $T_1 - T_2 = 50°C$ is equivalent to about $10\,Wm^{-2}$ when applying the long-wave radiation change $\epsilon\sigma \cdot 0.5(T_1^4 + T_2^4) - \epsilon\sigma \cdot (0.5 \cdot (T_1 + T_2))^4$ for typical temperatures on the Earth. Please note that the number $10\,Wm^{-2}$ is equivalent to a greenhouse gas forcing of more than quadrupling the $CO_2$ concentration in the atmosphere.~~

 The larger the seasonal contrast, the colder is the climate. Let us define here $\bar{}$ as the averaging over a time period (here the seasonal cycle), then $\overline{T^4} > \overline{T}^4$ which is consistent with Hölder's inequality (Rodgers, 1888; Hölder 1889; Hardy  1934, Kuptsov, 2001). It is noted that this feature is missing in the linearized version $A + B \cdot T'$ of the outgoing radiation. We see the large variation in the seasonal cycle $\Delta T = T_{summer} - T_{winter}$ for the blue line in Fig. **??** as compared to the dashed line. A mean change in the net long wave radiation can be approximated by the mean of summer and winter values $\epsilon\sigma \cdot 0.5(T_{summer}^4 + T_{winter}^4)$, which is up to $10\,Wm^{-2}$ higher than $\epsilon\sigma \cdot (0.5 \cdot (T_{summer} + T_{winter}))^4$ if the seasonal cycle is damped as in the dashed line of Fig. **??**. This implies that a lower seasonal cycle provides for a significant warming. If we would consider a linear model $A + B \cdot T'$ with $T'$ being measured in $°C$ for the long-wave radiation, the differences between the blue and the dashed line would be much lower, due to the absence of the non-linearity in net long wave radiation change.

**Comment 2)**

The second point relates to the vertical mixing in the ocean. It is interesting to see how the vertical mixing in the ocean obviously can affect the equator-to-pole surface temperature gradient. However, why should the vertical mixing be so different in the Palaeogene/Neogene before 3 Ma? Tidal dissipation can play a role, but also bathymetry and probably also the number and specific geometry of the ocean gateways. But so far, this remains very speculative and unmotivated in the manuscript. For example, how does the factor 25 in the vertical mixing coefficient that is used in the GCM simulations relate to expected changes in vertical mixing due to tides and bathymetry?

**Answer/Action**

The manuscript is admittedly a little vague at this point. A more explicit statement about a more explicit calculation of the vertical mixing is beyond the scope of the present paper. I will add more literature dealing with bathymetry, tides and geometry

of the ocean in the revised version. stressed out that The effective heat capacity is not an intrinsic property of the climate system but is reflective of the rate of penetration of heat energy into the ocean in response to the particular pattern of forcing and the background state (Schwartz, 2007). I mentioned the relevance for climate warming scenarios.

However, I have boiled down this chapter, rewrote most of it, and shortened it considerably. The factor of 25 is arbitrary. I wrote

In order to mimick the effect of a higher effective heat capacity and deepened mixed layer depth, the vertical mixing coefficient is increased in the ocean, changing the values for the background vertical diffusivity (arbritarily) by a factor of 25, providing a deeper thermocline.

and later

Furthermore, the model indicates that the respective winter signal of high-latitude warming is most pronounced (Fig. 9), decreasing the seasonality, suggesting a common signal of pronounced warming and weaker seasonality, a feature already seen in our EBM (Fig. 8).

and

Inspired by the EBM and GCM results, we may think of a climate system having a higher effective heat capacity producing a reduced seasonal cycle and flat temperature gradients. The changed vertical mixing coefficients are mimicking possible effects like weak tidal dissipation or abyssal stratification (e.g., Lambeck 1977; Green and Huber, 2013), but its explicit physics is not evaluated here.

Therefore, I deleted some text which was formally in there, given several arguments for expected changes in vertical mixing due to tides and bathymetry.

**Answer to interactive comment of referee #2 on "Temperatures from Energy Balance Models: the effective heat capacity matters"**

Gerrit Lohmann[1,2]

[1]Alfred Wegener Institute, Helmholtz Centre for Polar and Marine Research, Bremerhaven, Germany
[2] University of Bremen, Bremen, Germany

Thanks for the detailed critics on the *Interactive comment on Earth Syst. Dynam. Discuss., https://doi.org/10.5194/esd-2019-35, 2019.* by referee #2. The comments were helpful in improving the readability of the manuscript. In the following, I repeat and answer to these comments. Furthermore, the actions are described.

**Comment 1**

5  Eq. (4)

$$\epsilon \sigma T^4(\Theta, \varphi) = (1-\alpha)S \quad \cos\varphi \quad \cos\Theta \quad \times \quad 1_{[-\pi/2 < \Theta < \pi/2]}(\Theta)$$

is not a good description of the energy balance on the surface of the earth. A description of the spatial distribution of energy requires an introduction of energy redistribution by ocean currents, eddies, etc. Further, sun?s declination angle should be taken into account in order to describe reasonable spatial distribution of energy at any specific time of the year. What is described

10  here is, at best, an energy balance in an annual-mean sense when there is no physical mechanism for redistribution of strong energy surplus in the equatorial region and strong energy deficit in the polar region.

**Answer/Action**

The left-hand side of (4) as well as the right-hand side are latitude $\varphi$- and longitute $\Theta$-dependent. The incoming radiation goes with the cosine of latitude and longitude, and there is only sunshine during the day. This is noted as the $1_{[-\pi/2 < \Theta < \pi/2]}(\Theta)$

15  function which is zero outside the interval $[-\pi/2 < \Theta < \pi/2]$. The global mean temperature is not affected by the obliquity and precession (Berger and Loutre 1991; 1997; Laepple and Lohmann 2009). Therefore, we ignore the Earth orbital parameters for a while which makes an analytical calculation possible. Later in the numerical treatment, the full seasonal cycle is taken into account (former equation (16) in the manuscript). Indeed, eq. (4) is now better explained and motivated.

I have rewritten section 2, added a figure, and I think the notation is now clearer.

20  The incoming radiation goes with the cosine of latitude $\varphi$ and longitude $\Theta$, and there is only sunshine during the day. Fig. 1a shows the latitudinal dependence. As we assume no tilt (this assumption is later relaxed), the latitudinal dependence is a function of latitude only: $\cos\varphi$. On the right-hand side, the function is shown. Fig. 1b shows the latitudinal dependence is a function of longitude: $\cos\Theta$ for the sun-shining side of the Earth, and for the dark side of the Earth it is zero. For simplicity, we can define the angle $\Theta$ anti-clockwise on the for the sun-shining side between $-\pi/2$ and $\pi/2$. We define the maximal

[Figure]

**Figure 1.** New Figure 2 in the paper: Latitudinal (a) and longitudinal (b) dependence of the incoming short wave radiation. On the right hand side, the insolation as a function of latitude $\varphi$ and longitude $\Theta$ with maximum insolation $(1-\alpha)S$ is shown. See text for the details.

insolation always at $\Theta = 0$ which is moving in time. In the panel, the Earth's rotation is schematically sketched as the red arrow, and we see the time-dependence in the right-hand side. It is noted that the geographical longitude can be calculated by $mod(\Theta - 2\pi \cdot t/24, 2\pi)$ where t is measured in hours and mod is the modulo operation. Summarizing our geometrical considerations, we can now write the local energy balance as

5 $$\epsilon\sigma T^4 = (1-\alpha)S \cdot \cos\varphi \cdot \cos\Theta \qquad \text{for } -\pi/2 < \Theta < \pi/2 \tag{1}$$

and zero during night for $\Theta < -\pi/2$ or $\Theta > \pi/2$. Temperatures based on the local energy balance without a heat capacity would vary between $T_{min} = 0$ K and $T_{max} = \sqrt[4]{\frac{(1-\alpha)S}{\epsilon\sigma}} = \sqrt{2} \cdot \sqrt[4]{\frac{(1-\alpha)S}{4\epsilon\sigma}} = \sqrt{2} \cdot 288K = 407$ K.

Integration of (1) over the Earth surface is

$$\int_{-\pi/2}^{\pi/2} \left( \int_0^{2\pi} \epsilon \sigma T^4 R \cos\varphi \, d\Theta \right) R \, d\varphi \quad = \quad (1-\alpha) S \int_{-\pi/2}^{\pi/2} R \cos^2\varphi \, d\varphi \cdot \int_{-\pi/2}^{\pi/2} R \cos\Theta \, d\Theta$$

$$\epsilon \sigma R^2 \frac{4\pi}{4\pi} \int_{-\pi/2}^{\pi/2} \left( \int_0^{2\pi} T^4 \cos\varphi \, d\Theta \right) d\varphi \quad = \quad (1-\alpha) S R^2 \underbrace{\int_{-\pi/2}^{\pi/2} \cos^2\varphi \, d\varphi}_{\frac{\pi}{2}} \cdot \underbrace{\int_{-\pi/2}^{\pi/2} \cos\Theta \, d\Theta}_{2}$$

$$\epsilon \sigma 4\pi \overline{T^4} \quad = \quad (1-\alpha) S \, \pi \tag{2}$$

5  giving a similar formula as (3, in the paper) with the definition for the average $\overline{T^4}$.

What we really want is the mean of the temperature $\overline{T}$. Therefore, we take the fourth root of (1):

$$T = \sqrt[4]{\frac{(1-\alpha) S \cos\varphi \cos\Theta}{\epsilon \sigma}} \qquad \text{for } -\pi/2 < \Theta < \pi/2 \tag{3}$$

and zero elsewhere. If we calculate the zonal mean of (3) by integration at the latitudinal cycles we have

$$
\begin{aligned}
T(\varphi) \quad &= \quad \frac{1}{2\pi} \int_{-\pi/2}^{-\pi/2} \sqrt[4]{\frac{(1-\alpha) S \cos\varphi \cos\Theta}{\epsilon\sigma}} \, d\Theta \\[2mm]
&= \quad \frac{\sqrt{2}}{2\pi} \underbrace{\int_{-\pi/2}^{\pi/2} (\cos\Theta)^{1/4} d\Theta}_{\sqrt{\pi}\Gamma(5/8)/\Gamma(9/8)} \sqrt[4]{\frac{(1-\alpha)S}{4\epsilon\sigma}} \, (\cos\varphi)^{1/4} \quad = \quad \underbrace{\frac{1}{\sqrt{2\pi}} \frac{\Gamma(5/8)}{\Gamma(9/8)}}_{\approx 0.608} \cdot \sqrt[4]{\frac{(1-\alpha)S}{4\epsilon\sigma}} \, (\cos\varphi)^{1/4}
\end{aligned}
\tag{4}
$$

as a function on latitude (Fig. 3 in the paper). $\Gamma$ is Euler's Gamma function with $\Gamma(x+1) = x\Gamma(x)$. When we integrate this over the latitudes, we obtain

$$
\begin{aligned}
\overline{T} \quad &= \quad \frac{1}{2} \int_{-\pi/2}^{\pi/2} T(\varphi) \cos\varphi \, d\varphi \quad = \quad \frac{1}{2} \frac{\Gamma(5/8)}{\sqrt{2\pi}\Gamma(9/8)} \cdot \sqrt[4]{\frac{(1-\alpha)S}{4\epsilon\sigma}} \underbrace{\int_{-\pi/2}^{\pi/2} (\cos\varphi)^{5/4} d\varphi}_{\sqrt{\pi}\Gamma(9/8)/\Gamma(13/8)} \\[2mm]
&= \quad \frac{1}{2} \frac{1}{\sqrt{2}} \frac{\Gamma(5/8)}{\Gamma(13/8)} \cdot \sqrt[4]{\frac{(1-\alpha)S}{4\epsilon\sigma}} = \underbrace{\frac{\sqrt{2}}{4} \frac{8}{5}}_{0.4\sqrt{2}\approx 0.566} \cdot \sqrt[4]{\frac{(1-\alpha)S}{4\epsilon\sigma}}
\end{aligned}
\tag{5}
$$

15  Therefore, the mean temperature is a factor $0.4\sqrt{2} \approx 0.566$ lower than $288$ K as stated at (**??**) and would be $\overline{T} = 163$ K. The standard EBM in Fig. 1 (in the paper) has imprinted into our thoughts and lectures. We should therefore be careful and pinpoint the reasons for the failure. What happens here is that the heat capacity of the Earth is neglected and there is a strong non-linearity of the outgoing radiation.

**Comment 2**

Eq. (8): This is a strange derivation. Let us consider outgoing longwave radiation and incoming solar radiation in the form

$$\epsilon\sigma T^4(\phi,\Theta) \quad = \quad (1-\alpha)S \quad \cos\phi \quad \cos\Theta \quad \times \quad 1_{[-\pi/2<\Theta<\pi/2]}(\phi)$$

where $\phi$ is longitute and $\Theta$ is latitude. Equation (1) defines energy per unit time per unit area as the dimension of $\sigma = 5.670373 \times 10^{-8} Wm^{-2}K^{-4}$ indicates. Thus, total incoming radiation can be written as ...

$$(1-\alpha)S \pi R^2$$

Similarly, total outgoing radiation can be written as ...

$$\epsilon\sigma\overline{T}^4 4\pi R^2$$

Thus, we arrive at

5 $\quad \overline{T} = \sqrt[4]{\dfrac{(1-\alpha)S}{4\epsilon\sigma}}$

**Answer/Action**

Your equation above (in your review eq. (1)), is basically the same as the one I used. The difference is that you defined $\phi$ as longitude and $\Theta$ as latitude, whereas I do it in the other way round. (By the way: This is the same approach you criticized in your comment #1.)

10 $\quad$ The calculation you presented is also more or less the same as mine, with one **fundamental** difference. In my derivation by integration over the Earth surface

$\epsilon\sigma4\pi\overline{T^4} = (1-\alpha)S \pi$ .

In this formula, the average $\overline{T^4}$ is calculated, not $\overline{T}^4$. Therefore,

$\overline{T} = \sqrt[4]{\dfrac{(1-\alpha)S}{4\epsilon\sigma}}$

15 $\quad$ is not correct. This argument to exchange the $\overline{T^4}$ by $\overline{T}^4$ was one of the motivation to write down the global energy balance in a correct form. In order to $\overline{T}$, one has a more lengthy calculation: If we now calculate the zonal mean of the temperature by

integration at the latitudinal cycles we have

$$
\begin{aligned}
T(\varphi) &= \frac{1}{2\pi} \int\limits_{-\pi/2}^{-\pi/2} \sqrt[4]{\frac{(1-\alpha)S\cos\varphi\cos\Theta}{\epsilon\sigma}}\,d\Theta \\[2mm]
&= \frac{\sqrt{2}}{2\pi} \underbrace{\int\limits_{-\pi/2}^{\pi/2} (\cos\Theta)^{1/4}d\Theta}_{=\sqrt{\pi}\Gamma(5/8)/\Gamma(9/8)} \sqrt[4]{\frac{(1-\alpha)S}{4\epsilon\sigma}}\ (\cos\varphi)^{1/4} \\[2mm]
&= \frac{1}{\sqrt{2\pi}} \frac{\Gamma(5/8)}{\Gamma(9/8)} \cdot \sqrt[4]{\frac{(1-\alpha)S}{4\epsilon\sigma}}\ (\cos\varphi)^{1/4} \\[2mm]
&= 0.608 \cdot \sqrt[4]{\frac{(1-\alpha)S}{4\epsilon\sigma}}\ (\cos\varphi)^{1/4}
\end{aligned}
$$

as a function of latitude. $\Gamma$ is Euler's Gamma function with $\Gamma(x+1) = x\Gamma(x)$. When we integrate this over the latitudes, we obtain

$$
\begin{aligned}
\overline{T} &= \frac{1}{2} \int\limits_{-\pi/2}^{\pi/2} T(\varphi)\cos\varphi\,d\varphi \\[2mm]
&= \frac{1}{2} \frac{\Gamma(5/8)}{\sqrt{2\pi}\,\Gamma(9/8)} \cdot \sqrt[4]{\frac{(1-\alpha)S}{4\epsilon\sigma}} \underbrace{\int\limits_{-\pi/2}^{\pi/2} (\cos\varphi)^{5/4}d\varphi}_{=\sqrt{\pi}\Gamma(9/8)/\Gamma(13/8)} \\[2mm]
&= \frac{1}{2}\frac{1}{\sqrt{2}}\frac{\Gamma(5/8)}{\Gamma(13/8)}\sqrt[4]{\frac{(1-\alpha)S}{4\epsilon\sigma}} = \frac{\sqrt{2}}{4}\frac{8}{5}\sqrt[4]{\frac{(1-\alpha)S}{4\epsilon\sigma}} \\[2mm]
&= 0.4\sqrt{2}\sqrt[4]{\frac{(1-\alpha)S}{4\epsilon\sigma}} = 0.566\sqrt[4]{\frac{(1-\alpha)S}{4\epsilon\sigma}}
\end{aligned}
$$

As a side remark: The fact that $\sqrt[4]{\overline{T^4}}$ is higher than $\overline{T}$ is consistent with Hölder's inequality (Rodgers, 1888; Hölder 1889; Hardy et al., 1934, Kuptsov, 2001). This now mentioned in the revised version of the manuscript.

**Comment 3**

Specific heat is not needed to reproduce the reasonable spatial distribution of temperature. For example, 1D energy balance model with meridional heat flux can be written as (see North and Kim, 2017; p123-134)

$$
-\frac{d}{d\mu}\left(D(1-\mu^2)\frac{dT(\mu)}{d\mu}\right) + A + BT(\mu,t) = Qa(\mu)s(\mu)
$$

. . .

with a realistic insolation distribution function and a realistic albedo (see North and Kim, 2017 for details), we obtain a solution as in Figure 1. The model solution is fairly similar to the observational data. Further, the diffusive heat transport in a zonal mean sense looks very reasonable compared to that derived from satellite observations (see Figure 2).

**Answer/Action**

Thanks for your comment and hinting to the important work of North and Kim (2017). As I mention in the manuscript: "The linearization of the long wave radiation in several models (North et al., 1975a, b; Chen et al., 1995) implicitly assumes the above heat capacity and fast rotation arguments. " Indeed, the linearized version can give a reasonable zonal mean climate.

5    In the revised version, I stress out this more clearly. My point is that we need a rapidly rotating object with significant heat capacity. Without these effects, the global mean temperature would be lower.

I inserted the text: Quite often the linearization the long wave radiation $\epsilon\sigma T^4$ is linearized in energy balance models. Indeed the linearization is performed around $0°C$ (North et al., 1975a, b; Chen et al., 1995; Lohmann and Gerdes, 1998; North and Kim, 2017) and is formulated as $A + B \cdot T'$ with $T'$ being measured in $°C$. As the temperatures based on the local energy

10    balance without a heat capacity would vary between $T_{min} = 0$ K and $T_{max} = \sqrt{2} \cdot 288K = 407$ K, a linearization would be not permitted. Therefore, the linearization implicitely assumes the above heat capacity and fast rotation arguments.

I show that other processes - like horizontal transport processes - are only of secondary importance for the globally averaged temperature (Figs. 3 and 6) The finding is furthermore important to see that the effective heat capacity (which is time-scale dependent) has a direct influence on the global (and regional) temperatures (Figs. 4 and 5). Fig. 5 shows how the temperature

15    would change in a slowly rotating planet.

**Comment 4**

"The atmospheric circulation provides an efficient way to propagate heat along latitudes which is ignored and is a second order effect (not shown)." This statement is erroneous. As demonstrated in Comment #3 above, a reasonable temperature distribution on the surface of the earth is reproduced by using diffusive heat transport. Heat capacity is not even used in this

20    calculation of equilibrium temperature.

**Answer/Action**

Yes, this can be better motivated. Indeed, the energy input is time-dependent. Therefore, the mean and latitudinal temperature depends on the time derivative. See also my comments and action points in answer to Comment #3. As seen in the manuscript, the temperatures do depend on the effective heat capacity.

25    In the revised version, I am clearer is stating in the abstract

Energy balance models (EBM) are highly simplified systems of the climate system. The global temperature is calculated by the radiation budget through the incoming energy from the Sun and the outgoing energy from the Earth. The argument that the temperature can be calculated by the simple radiation budget is revisited. The underlying assumption for a realistic temperature distribution is explored: One has to assume a moderate diurnal cycle due to the large heat capacity and the fast rotation of the

30    Earth. Interestingly, the global mean in the revised EBM is very close to the originally proposed value.

**Comment 5**

Eq. (12): The author introduced diurnal cycle of temperature and determined the global average of the averaged diurnal cycle of temperature. This discussion is erroneous. We can write diurnal temperature change as ...

Further, incoming solar radiation has the same form as in (1). Thus, we arrive at the same conclusion as in Comment #2.

**Answer/Action**

Thanks for making this hint. Indeed, I agree to your basic equation (1), but not with the conclusion. See my answer to your Comment #2.

**Comment 6**

A time-dependent 1D EBM can be written as ...

Equation (12) is equivalent to (6) as far as the annual mean temperature is concerned. Obviously, an adequate explanation is needed in terms of how Figure 3 is produced.

**Answer/Action**

I agree for the linearized EBM. See my answer to your Comment #3. I explicitly state that in the linearized EBM, the heat capacity argument is implicitly included. in the abstract: A linearized EBM implicitly assumes the heat capacity and the fast rotation arguments.

and

Quite often the linearization the long wave radiation $\epsilon\sigma T^4$ is linearized in energy balance models. Indeed the linearization is performed around $0°C$ (North et al., 1975a, b; Chen et al., 1995; Lohmann and Gerdes, 1998; North and Kim, 2017) and is formulated as $A + B \cdot T'$ with $T'$ being measured in $°C$. As the temperatures based on the local energy balance without a heat capacity would vary between $T_{min} = 0$ K and $T_{max} = \sqrt{2} \cdot 288K = 407$ K, a linearization would be not permitted. Therefore, the linearization implicitely assumes the above heat capacity and fast rotation arguments.

Furthermore, parameter study of Figure 3 is better explained. The model that I use is the non-linear model with the $T^4$-term and a time-dependent forcing. I insert furthermore Fig. 4 with the temperature dependence on heat capacity (and rotation rate) when analyzing the diurnal cycle.

**Comment 7**

Figure 3: The main effect of heat capacity in the original EBM is in the context of the amplitude of the annual and semi-annual cycles (see North and Kim, p152). The annual and semi-annual cycles are seriously affected by the choice of heat capacity, whereas the annual mean component is not. The author should demonstrate that not only annual-mean temperature distribution but also the annual and semi-annual cycles of temperature is reproduced reasonably by their choice of heat capacity (see, for example, Fig. 6.8 of North and Kim).

**Answer/Action**

Thanks again for your comment and for hinting at the important work of North and Kim (2017). Yes, I agree for the linearized EBM. See my answer to your Comment #3 and #6. In the non-linear model, the temperature is affected by the heat capacity. The effective heat capacity is important as it is reflective of the rate of penetration of heat energy into the ocean in response to the particular pattern of forcing and the background state (Schwartz, 2007). I also emphasize the relevance of this for climate scenarios.

In section 5, I write now For the diurnal cycle $h_T$ is less that half a meter and the heat capacity generally less than that of the atmosphere. The seasonal mixed layer depth can be several hundred meters (e.g., de Boyer Montégut et al., 2004). As pointed out by Schwartz (2007), the effective heat capacity that reflects only that portion of the global heat capacity that is coupled to the perturbation on the timescale of the perturbation. In the context of global climate change induced by changes in atmospheric composition on the decade to century timescale the effective heat capacity is subject to change in heat content on such timescales.

I explicitly show the results of the linearized EBM here. In principle there is no principle difference to the non-linear model, although in detail the distribution is different.

In section 4, I now explicitly write Fig. 8 shows the seasonal amplitude for the $C_p$-scenarios as indicated by the blue and dashed black lines, respectively. The larger the seasonal contrast, the colder is the climate. Let us define here $\bar{\ }$ as the averaging over a time period (here the seasonal cycle), then $\overline{T^4} > \overline{T}^4$ which is consistent with Hölder's inequality (Rodgers, 1888; Hölder 1889; Hardy et al., 1934, Kuptsov, 2001). It is noted that this feature is missing in the linearized version $A + B \cdot T'$ of the outgoing radiation. We see the large variation in the seasonal cycle $\Delta T = T_{summer} - T_{winter}$ for the blue line in Fig. **??** as compared to the dashed line. A mean change in the net long wave radiation can be approximated by the mean of summer and winter values $\epsilon\sigma \cdot 0.5(T_{summer}^4 + T_{winter}^4)$, which is up to $10\,Wm^{-2}$ higher than $\epsilon\sigma \cdot (0.5 \cdot (T_{summer} + T_{winter}))^4$ if the seasonal cycle is damped as in the dashed line of Fig. 8. This implies that a lower seasonal cycle provides for a significant warming. If we would consider a linear model $A + B \cdot T'$ with $T'$ being measured in $^\circ C$ for the long-wave radiation, the differences between the blue and the dashed line would be much lower, due to the absence of the non-linearity in net long wave radiation change.

This clarifies the difference between a linear and non-linear model.

[Figure]

**Figure 2.** Equilibrium temperature of the linearized EBM using different diffusion coefficients. ). Units are °C.

**Comment 8**

P8 L20: What does the first law of thermodynamics have anything to do with incoming radiation = outgoing radiation? Is the author referring to the zeroth law of thermodynamics?

**Answer/Action**

Thanks. This is indeed not well formulated. In the revised version, I deleted the phrase $1^{st}$ law of thermodynamics on the

**Comment 9**

As already demonstrated in Comment #2, global average temperature is close to the observed value without the effect of heat capacity. Further, by using diffusive heat transport, zonally average temperature is reproduced close to actual observation in Comment #3.

**Answer/Action**

Thanks. I disagree and point to my answers to Comments #2 and #3.

**Comment 10**

It is difficult to review the entire manuscript until my earlier comments are fully addressed. In particular, the author needs to explain clearly how the solutions in each figure are computed (with appropriate equation if possible) and demonstrate clearly that his full solution (with diurnal and annual cycles) matches reasonably with the observations for his choice of heat capacity.

**Answer/Action**

I agree that it seems a misunderstanding. My answers to Comments #2 and #3 shall clarify the mistake in the calculation of the temperature. The time dependence is more explicitly stated. Furthermore, the non-linearity in the outgoing radiation makes this model different from your EBMs. The equations for the EBMs used are in the manuscript. For the equations, it is important to explicitly spell out the assumptions made. See my answer to your Comments #3 and #6. It is now explicitly stated that in the linearized EBM, the heat capacity argument is implicitly included. The model that I use is the non-linear model with the $T^4$-term and a time-dependent forcing.

Furthermore, I refer to the results of the linearized EBM in the revised version. Indeed, the linear model has some advantages because it directly indicated the climate sensitivity. However, for a rigorous derivation of the global and regional effects, the non-linear version is an advantage. Finally, it is necessary to see that $\overline{T^4}$ is not $\overline{T}^4$. Simplified and conceptual models can be used to study long-term climate (Hasselmann, 1976; Lemke, 1977; Timmermann and Lohmann, 2000; Lohmann, 2018). As pointed out in the manuscript, the effective heat capacity is important to understand past and potential future climate.

**Technical Comment 1**

There is, in general, lack of explanation for variables used in the equations.

**Answer/Action**

Thanks. I have gone through all equations in the manuscript to avoid misunderstandings.

**Technical Comment 2**

5    "shown in Fig. 2 as the (read) red line with the mean ?"

**Answer/Action**

Thanks, corrected.

**Answer to interactive comment of Referee #3 on "Temperatures from Energy Balance Models: the effective heat capacity matters"**

Gerrit Lohmann[1,2]

[1]Alfred Wegener Institute, Helmholtz Centre for Polar and Marine Research, Bremerhaven, Germany
[2] University of Bremen, Bremen, Germany

Thanks for the constructive critics in the *Interactive comment on Earth Syst. Dynam. Discuss., https://doi.org/10.5194/esd-2019-35, 2019.* by referee #3. In the following, I repeat and answer to these comments. Furthermore, the actions are documented.

**Comment 1**

5  Neglecting the diurnal cycle in EBMs is a rather standard procedure. This assumes that the Earth receives a mean daily incoming solar energy equally distributed over each latitude bands. This is indeed most of the time quite a reasonable hypothesis for such simplified models, since the ocean surface temperature diurnal changes are small (at most a few degrees). This paper confirms this usual assumption, with the red and dotted brownish curves of Figure 2 being almost indistinguishable.

The presentation on this section is however extremely confusing. The author starts with the classical 0-dimensional time
10  average EBM. He then presents the 1-dimensional case with a daily cycle as an extension, just introducing it as a local extension of the 0-dimensional case. However, considering that there is a local energy balance is not a valid assumption in general, contrary to the one at the global scale. Obviously, it is clearly entirely irrelevant to consider that there can be an instantaneous radiative equilibrium, with temperatures dropping to zero Kelvin as soon as the Sun sets. This is clearly not what people usually assume when using EBMs !

15  The usual starting point corresponds to equations (11-12) where the solar forcing is averaged over one Earth rotation. This is more or less what people have been using in geographically explicit EBMs, including the very first ones. Budyko and Sellers 1969 where indeed geographically explicit, without a diurnal cycle, as in equation (11). The authors comes back to it as a compensation of the incoherent assumption of a local radiative equilibrium. So part 2 is just showing that an irrelevant hypothesis produces irrelevant results. It brings nothing interesting, but only confusion.

20  To add to the confusion some assumptions are clearly not explained. On the top of page 4, the first equation is clearly invalid unless strong hypotheses are imposed, which are not specified in the text.

**Answer/Action**

The confusion shall be clarified. This manuscript revisits the relationship between the (global mean) surface temperature of the Earth and its radiation budget as is frequently used in Energy balance models (EBMs). The main point is, that the effective

heat capacity (and its temporal variation over the daily/seasonal cycle) needs to be taken into account when estimating surface temperature from the energy budget. As a starting point, a zero-dimensional model of the radiative equilibrium of the Earth is introduced

$$(1-\alpha)S\pi R^2 = 4\pi R^2 \epsilon\sigma T^4 \tag{1}$$

where the left-hand side represents the incoming energy from the Sun while the right-hand side represents the outgoing energy from the Earth. This is used to calculate the temperature

$$T = \sqrt[4]{\frac{(1-\alpha)S}{4\epsilon\sigma}} \tag{2}$$

The wording "This is clearly not what people usually assume when using EBMs" shows that there are implicit assumptions in the approach. To my point of view, the assumptions can be explicitly spelled out to obtain arguments which steps are necessary to make. I show that the global energy balance should not be calculated from this approach, because it neglects the implicit assumption of a fast rotating Earth with significant heat capacity. I am not aware of a paper which explicitly shows that

$$T = 0.989 \sqrt[4]{\frac{(1-\alpha)S}{4\epsilon\sigma}} \quad . \tag{3}$$

The author knows the fundamental work of Budyko (1969) and Sellers (1969) where the EBM could be geographically explicit, but their result has not be used to calculate the mean temperature (3).

Your statement that the author comes back to the geographically explicit EBM as a compensation of the incoherent assumption of a local radiative equilibrium cannot be found in the manuscript. The calculation of the global mean temperature from the energy balance is not irrelevant.

Your final point is that the first equation on the top of page 4 is clearly invalid unless strong hypotheses are imposed, which are not specified in the text. If you see the text above this equation, it is clearly written that it is about the diurnal variation. In a revised version, I explicitly spell out that this approximation is exactly the point of the low diurnal cycle due to the heat capacity.

I have rewritten section 2, added a figure, and I think the notation is now clearer.

[revised manuscript text omitted]

**Comment 2**

The second part of the paper discusses the role of heat capacity in the ?diurnal averaging? of temperatures. Results are summarized on Fig.3. As discussed above, the fact that temperatures are much lower for small heat capacities is rather obvious (with Earth losing most of, or all its thermal energy during the night).

**Answer/Action**

I am not aware of a study analyzing the effect of the heat capacity on global climate. Indeed, the manuscript shows this effect in Fig. 4 of the revised paper. The effective heat capacity is time-scale dependent. For the day and night cycle values in the order of the atmospheric heat capacity are realistic for our Earth with 24 h rotation. In the revised version, I am more explicit here and show how the temperature would change in a slowly rotating planet (new Fig. 5). The heat capacity plays also a role when dealing with the seasonal cycle (Figs. 7 and 8).

**Comment 3**

Using the typical oceanic vertical diffusivities for estimating a heat capacity is not very relevant. The diurnal cycle is buffered by the very top layers of the ocean that are usually almost well-mixed by winds and also by the diurnal cycle itself. The interior ocean vertical diffusivity has no role.

**Answer/Action**

The effective heat capacity is indeed time-scale dependent. In section 5, I write now For the diurnal cycle $h_T$ is less that half a meter and the heat capacity generally less than that of the atmosphere. The seasonal mixed layer depth can be several hundred meters (e.g., de Boyer Montégut et al., 2004). As pointed out by Schwartz (2007), the effective heat capacity that reflects only that portion of the global heat capacity that is coupled to the perturbation on the timescale of the perturbation. In the context of global climate change induced by changes in atmospheric composition on the decade to century timescale the effective heat capacity is subject to change in heat content on such timescales.

**Comment 4**

I do not see what is the purpose of solving equation (15) and showing Figure 4. This does not relate to the diurnal cycle, nor to heat capacity, nor to vertical mixing. What is the point ? The statement "global mean temperature is not affected by the transport because of the boundary condition. . .." is a bit strange. I would write more simply that here, global mean temperature is a measure of global heat content (uniform heat capacity) which depends only of global net radiative fluxes, not internal redistribution.

**Answer/Action**

The purpose of equation (15) and former Fig. 4 was to show the influences of the meridional heat transport and the seasonal cycle. They do not change the global mean temperature, but the temperature gradient. The statement "global mean temperature is not affected by the transport because of the boundary condition" was written to show the reader that the heat transport does not play a role (unless other feedbacks are included). In the revised version, I adopted your formulation.

**Comment 5**

Bottom of page 5 "Until now we assumed that the Earth's axis ...": It is quite awkward to explain only now where equation (4) page 2 comes from. Indeed equation (4) is certainly not standard for the Earth in particular in the context of EBMs and climate modeling. A planet with no tilt has no seasonal cycle. Many EBMs have an explicit seasonal cycle. Again, the starting point of the paper is very awkward.

**Answer/Action**

At the bottom of page 5, it is not the first point where equation (4) comes from. Equation (4) comes directly from the basic incoming radiation. I re-wrote the paragraph (see your comment 1). I see now the point that the motivation for equation (4) in the manuscript needs a better introduction. The beauty of (4) is that we ignore the Earth orbital parameters first (no obliquity

and no precession) which makes an analytical calculation possible. The global mean temperatures are not affected by the tilt
and the values are identical to the one calculated. Indeed is stated earlier in the manuscript.

Let us have a closer look onto (1) and consider local radiative equilibrium of the Earth at each point. Fig. 1 shows the latitude-longitude dependence of the incoming short wave radiation. The global mean temperatures are not affected by the tilt (Berger and Loutre 1991; 1997; Laepple and Lohmann 2009). We assume an idealized geometry of the Earth, no obliquity and no precession, which makes an analytical calculation possible.

**Comment 6**

In the last part, the author presents some experiments with the COSMOS coupled model, to investigate the role of vertical mixing on the meridional temperature gradient. Unfortunately, it is not clear at all that these results are linked to the diurnal cycle or heat capacity. The author sets an experiment with an 25-fold increase in the background diffusivity. The logical outcome of this experiment should be to increase dramatically the oceanic circulation (not shown in the manuscript) and thus to increase massively the heat transport and the vertical mixing in the ocean. How does this relate to the heat capacity or the diurnal cycle is a mystery for me and how conclusions can be drawn from there is likewise impossible to understand. The only clear result is a weakening of the equator-to-pole gradient (likely due to increase heat transport by the ocean); however there is no physical basis to link this to past climates as the author is doing, since no probable mechanism can be suggested to increase the diffusivity by a factor 25 globally.

**Answer/Action**

Energy balance models have been used to diagnose the temperatures on the Earth when applying complex circulation models. The outcome of the new approach is that effective heat capacity matters for the climate system. This cannot be seen in

$$T = \sqrt[4]{\frac{(1-\alpha)S}{4\epsilon\sigma}} \tag{9}$$

The motivation is that we may think of a climate system having a higher net heat capacity producing flat temperature gradients and reduced seasonal cycle. I replaced this section by

Fig. 8 shows the seasonal amplitude for the $C_p$-scenarios as indicated by the blue and dashed black lines, respectively. A change in the seasonal /diurnal cycle of $T_1 - T_2 = 50°C$ is equivalent to about $10 Wm^{-2}$ when applying the long wave radiation change $\epsilon\sigma \cdot 0.5(T_1^4 + T_2^4) - \epsilon\sigma \cdot (0.5 \cdot (T_1 + T_2))^4$ for typical temperatures on the Earth. Please note that the number $10 Wm^{-2}$ is equivalent to a greenhouse gas forcing of more than quadrupling the $CO_2$ concentration in the atmosphere.

Energy balance models have been used to diagnose the temperatures on the Earth when applying complex circulation models (e.g., Knorr et al. 2011)or data (e.g., Köhler et al., 2010; van der Heydt et al., 2016; Stap The larger the seasonal contrast, the colder is the climate. Let us define here $\bar{\cdot}$ as the averaging over a time period (here the seasonal cycle), then $\overline{T^4} > \overline{T}^4$ which is consistent with Hölder's inequality (Rodgers, 1888; Hölder 1889; Hardy et al., 2018). For the past, a strong warming at high latitudes is reconstructed for the Pliocene, Miocene, Eocene periods (Markwick, 1994; Wolfe, 1994; Sloan and Rea, 1996; Huber et al. , 2000; Shellito et al. , 2003; Tripati et al. , 2003; Mosbrugger et al., 2005; Utescher and Mosbrugger, 2007). 1934, Kuptsov, 2001). It is noted that this feature is missing in the linearized version $A + B \cdot T'$ of the outgoing radiation.

We see the large variation in the seasonal cycle $\Delta T = T_{summer} - T_{winter}$ for the blue line in Fig. **??** as compared to the dashed line. A mean change in the net long wave radiation can be approximated by the mean of summer and winter values $\epsilon\sigma \cdot 0.5(T_{summer}^4 + T_{winter}^4)$, which is up to $10\,Wm^{-2}$ higher than $\epsilon\sigma \cdot (0.5 \cdot (T_{summer} + T_{winter}))^4$ if the seasonal cycle is damped as in the dashed line of Fig. **??**. This implies that a lower seasonal cycle provides for a significant warming. If we would consider a linear model $A + B \cdot T'$ with $T'$ being measured in $^\circ C$ for the long-wave radiation, the differences between the blue and the dashed line would be much lower, due to the absence of the non-linearity in net long wave radiation change.

I rewrote most of the section, shortened it and elaborated the link to the seasonal cycle. The aim was to show one of the potential consequences of the effective heat capacity to explore the full range of solutions. Indeed, the increase in the mixing is admittedly ad hoc. Therefore, I have boiled down this chapter, rewrote most of it, and shortened it considerably. I wrote

In order to mimic the effect of a higher effective heat capacity and deepened mixed layer depth, the vertical mixing coefficient is increased in the ocean, changing the values for the background vertical diffusivity (arbitrarily) by a factor of 25, providing a deeper thermocline.

and later

Furthermore, the model indicates that the respective winter signal of high-latitude warming is most pronounced (Fig. 9), decreasing the seasonality, suggesting a common signal of pronounced warming and weaker seasonality, a feature already seen in our EBM (Fig. 8).

and

Inspired by the EBM and GCM results, we may think of a climate system having a higher effective heat capacity producing a reduced seasonal cycle and flat temperature gradients. The changed vertical mixing coefficients are mimicking possible effects like weak tidal dissipation or abyssal stratification (e.g., Lambeck 1977; Green and Huber, 2013), but its explicit physics is not evaluated here.

**Comment 7**

In the conclusion, there are some mentions of possible linearization of the long wave radiation. Since this is critical to the whole paper (averaging T is not the same as averaging $T^4$ ), I am surprised not to see a much more detailed discussion of this point much earlier in the paper.

**Answer/Action**

In the revised version, I can stress out this more clearly. My point is that we need a rapidly rotating object with significant heat capacity. Without these effects, the global mean temperature would be lower. I explicitly state that in the linearized EBM, the heat capacity argument is implicitly included. in the abstract: A linearized EBM implicitly assumes the heat capacity and the fast rotation arguments.

and

Quite often the linearization the long wave radiation $\epsilon\sigma T^4$ is linearized in energy balance models. Indeed the linearization is

performed around $0°C$ (North et al., 1975a, b; Chen et al., 1995; Lohmann and Gerdes, 1998; North and Kim, 2017) and is

5  formulated as $A + B \cdot T'$ with $T'$ being measured in $°C$. As the temperatures based on the local energy balance without a heat capacity would vary between $T_{min} = 0$ K and $T_{max} = \sqrt{2} \cdot 288K = 407$ K, a linearization would be not permitted. Therefore, the linearization implicitely assumes the above heat capacity and fast rotation arguments.

In section 4, I now explicitly write Fig. 8 shows the seasonal amplitude for the $C_p$-scenarios as indicated by the blue and dashed black lines, respectively. The larger the seasonal contrast, the colder is the climate. Let us define here $\overline{\cdot}$ as the averaging

10  over a time period (here the seasonal cycle), then $\overline{T^4} > \overline{T}^4$ which is consistent with Hölder's inequality (Rodgers, 1888; Hölder 1889; Hardy et al., 1934, Kuptsov, 2001). It is noted that this feature is missing in the linearized version $A + B \cdot T'$ of the outgoing radiation. We see the large variation in the seasonal cycle $\Delta T = T_{summer} - T_{winter}$ for the blue line in Fig. **??** as compared to the dashed line. A mean change in the net long wave radiation can be approximated by the mean of summer and winter values $\epsilon\sigma \cdot 0.5(T^4_{summer} + T^4_{winter})$, which is up to $10\,Wm^{-2}$ higher than $\epsilon\sigma \cdot (0.5 \cdot (T_{summer} + T_{winter}))^4$ if the seasonal

15  cycle is damped as in the dashed line of Fig. 8. This implies that a lower seasonal cycle provides for a significant warming. If we would consider a linear model $A + B \cdot T'$ with $T'$ being measured in $°C$ for the long-wave radiation, the differences between the blue and the dashed line would be much lower, due to the absence of the non-linearity in net long wave radiation change.

This clarifies the difference between a linear and non-linear model.

---

## Editor Decision (ED1)

I think that the manuscript now is very close to "be accepted" status and required corrections are very minor indeed. But still …

1. I am still not completely satisfied with the explanations behind the derivation of (10). Yes, for observational data the diurnal cycle is weak and Fig.11 shows that $\overline{(T^4)} \cong (\overline{T})^4$ but what about the solution of (9) where this assumption is used? Would you please provide the estimate for the amplitude of diurnal cycle and analogue of Fig.11 obtained for numerical solution of model (9) at some reasonable value of the heat capacity.

2. Page 1, line 1 "….simplified systems of the climate system….". Please reword.

3. Page 4, line 9. I would also add that the "local radiative equilibrium assumption" cannot be used at these conditions.

3. Page 4, line 12 "elesewhere". Please correct.

4. Page 10, line 18. I would not use "Ironically" here. Two estimates are close because the Earth is a fast rotating body with significant heat capacity having spherical geometry. Original estimate implicitly uses these facts. I do not see any irony here.

---

## Author Response (AR2)

**Answer to interactive comment of Referee #1 (second round) on "Temperatures from Energy Balance Models: the effective heat capacity matters"**

Gerrit Lohmann[1,2]

[1]Alfred Wegener Institute, Helmholtz Centre for Polar and Marine Research, Bremerhaven, Germany
[2] University of Bremen, Bremen, Germany

Thanks for the constructive critics on the revised version of the manuscript *Earth Syst. Dynam. Discuss., https://doi.org/10.5194/esd-2019-35, 2019.* by referee #1. In the following, I repeat and answer to these comments (in blue). Furthermore, the actions are documented.

**Comment 1**

5    A reasonable range of heat capacity should be used and demonstrate that all aspects of meridional temperature profile are appropriately reproduced;

Answer: For the dynamics (9), a range of heat capacities were used to show the meridional temperature profiles. Figure 4 in the revised version is exactly showing a severe drop in temperatures for heat capacities below 0.01 of the atmospheric heat capacity $C_p^a$. The climate is relatively insensitive to changes in heat capacity $C_p \in [0.05 \cdot C_p^a, 2 \cdot C_p^a]$. This supported by Figure

10    5 in the revised version.

**Comment 2**

Compare the relative importance of heat capacity against other mechanisms (such as albedo feedback).

Answer: This is indeed an important point. When comparing the dashed and dotted blue lines in Figure 6 of the revised ver-

15    sion, the ice-albedo-feedback can cause a temperature drop at polar latitudes of $\sim 10$ K. At low latitudes, the effect is minor, and the global mean is affected by about 2-3 K. The effect of the heat capacity is about 125 K according to (8) for depending on $C_p$ (Figures 3 and 4). In the revised version, such comparison is emphasized in the conclusions section.

**Major Comments**

20    1) I think that the argument hinged around (7) and (8) is extremely futile. Nobody assumes that (4) is a valid equation on a short daily time scale. The Earth's atmosphere and ocean will not reach an equilibrium as described in (4) on a daily time scale.

Answer: This is exactly one of my main points. It is important to evaluate which assumption has been made in deriving the energy balance. The calculation in (7,8) shows that the heat capacity is necessary in the equation, showing that (9) is a better

description than the equation without the time derivative. In general, the heat capacity term cannot be ignored on the left hand side, since the right hand side is time-dependent. This is emphasized in the conclusions section.

2) Figure 4: Even after incorporating heat capacity in (9), the equilibrium daily average temperature in (11) does not contain heat capacity. Explain why solutions incorporating different values of Cp in Fig. 4 show different meridional structures. Why are numerical solutions different from the exact solution?

Answer: The main point is, that the effective heat capacity (and its temporal variation over the daily/seasonal cycle) needs to be taken into account when estimating surface temperature from the energy budget. The solution depends on the heat capacity (as indicated by Figure 4). The finding is consistent with a sensitivity study of climatological SST to slab ocean model thickness (Wang et al., 2019).

Let us define $\Delta T = T_{max} - T_{min}$ for $T_{max}$ and $T_{min}$ being the maximum and the minimum temperature during the diurnal or seasonal cycle. What I demonstrate is: the larger the daily/seasonal contrast, the colder is the climate. Let us define here $\overline{*}$ as the averaging over a time period, then $\overline{T^4} > \overline{T}^4$ which is consistent with Hölder's inequality (Rodgers, 1888; Hölder 1889; Hardy et al., 1934, Kuptsov, 2001).[1] We see a large variation in the seasonal cycle $\Delta T = T_{summer} - T_{winter}$ (blue line compared to the dashed line in Fig. 8: about 50 K for high latitudes) or $\Delta T = T_{day} - T_{night}$ (Figure 2b with variation between 407 and 0 K according to lines 17/18 on page 3).

If we approximate the mean change in the net long wave radiation by $\epsilon \sigma \cdot 0.5(T_{max}^4 + T_{min}^4)$, we see that is much higher than $\epsilon \sigma \cdot (0.5 \cdot (T_{max} + T_{min}))^4$. If the seasonal or daily cycles are damped, the outgoing radiation is smaller and contributes to a significant warmer climate.

The numerical solution (dashed brown in Figure 3) is close to the analytical solution (red line in Figure 3). I have to emphasize that the numerical solution takes directly (9), whereas the analytical approach takes the assumption in (10) into account. Therefore, the solutions are close, but not identical. This has nothing to do with numerical schemes etc. I explicitly wrote which curve is related to which approximation in section 2 of the new version.

3) P8 L9-34: It seems that the author is seeking a reason for the amplified high- latitude warming from deep mixed layer (or increased heat capacity). We already know that the seasonal cycle is flattened by using a large value of heat capacity. (That is why the amplitude of the seasonal cycle is very weak over the ocean than over the continent.) By using a large heat capacity, polar temperature in the winter hemisphere will rise significantly. It is not clear what the author is trying to prove or demonstrate here.

Answer: What I demonstrate is: the larger the seasonal contrast, the colder is the climate. This is a consequence of the non-linearity as explained above. This effect is not included in the linearized version. I emphasize this statement in the paper both for the seasonal and diurnal cycle.
* * *
[1] It is noted that this feature is missing in the linearized version $A + B \cdot T'$ of the outgoing radiation. $T'$ in °C.

4. P9 L15: "I argue ... " Unfortunately, the idea of using heat capacity is already well known. No one will assume that diurnal temperature variation is in successive thermal equilibrium.

Answer: I cannot see that the heat capacity is used in the literature. The arguments are

$$(1-\alpha)S = 4\epsilon\sigma T^4$$

Solving for the temperature,

$$T = \sqrt[4]{\frac{(1-\alpha)S}{4\epsilon\sigma}}$$

The wording "no one will assume" shows that there are implicit assumptions in the approach. To my point of view, the

5    assumptions can be explicitly spelled out to obtain arguments which steps are necessary to make. I show that the global energy balance should not be calculated from this approach, because it neglects the implicit assumption of a fast rotating Earth with significant heat capacity. I am not aware of a paper which explicitly shows that

$$T = G\sqrt[4]{\frac{(1-\alpha)S}{4\epsilon\sigma}} \quad \text{with } G = \sqrt{\frac{\pi}{2}}\frac{\Gamma(9/8)}{\Gamma(13/8)} \approx 0.989 \tag{1}$$

The author knows the fundamental work of Budyko (1969) and Sellers (1969) where the EBM could be geographically explicit,

10    but their result has not be used to calculate the mean temperature.

5. P9 L19: "The Earth system understanding ..." This statement, I believe, is not quite true. We already know that the diurnal temperature is not in equilibrium with the incoming solar radiation on a daily time scale. Therefore, the governing equation in (4) is unreasonable for depicting the diurnal temperature variation, and we have to add a time-dependent term in the governing

15    equation. If we use a linearized EBM, the diurnal cycle of temperature satisfies ...

$$\tilde{T} = \frac{1-\alpha}{\pi B}S\cos\varphi - \frac{A}{B}$$

be seen in Fig. R1, however, the meridional gradient of temperature is too steep to be realistic. In order to have a reasonable meridional temperature profile, heat transport term should be included in the energy balance model. Note further that global average temperature and meridonal temperature gradient cannot simultaneously be tuned by using heat capacity. Also, the author should use a reasonable range of heat capacity to address its importance.

20    Answer: I would like to clarify the issue. a) The linearized model has its value and I used it also in previous studies. However, the implicit assumption is the averaging over time. If we linearize, we shall assume that the temperature is close to an equilibrium around which the linearization can be taken. This is not the case in your equation (1). One shall also specify the units of temperature. $A + BT$ is the linearized version of the long wave radiation around $0°$C.

b) There is a misunderstanding. The global average temperature and meridional temperature gradient are not tuned by using

25    the heat capacity. The global averaged temperature is independent on the heat transport. The basic assumption is that the meridional heat transport vanishes at the poles.

I stated in the text: "The global mean temperature is not affected by the transport term because it depends 10 only of global net radiative fluxes, not internal redistribution. Formally, the integration with boundary condition with zero heat transport at the poles provides no effect." Formally, $HT \sim \partial_y T = 0$ at the poles ($\pm\pi/2$). Therefore

$$\int\limits_{-\pi/2}^{\pi/2} dy\, \partial_y^2 T = \partial_y T|_{-\pi/2}^{\pi/2} = 0$$

showing that the heat transport has no effect on the global mean temperature. In section 4, page 6, line 11, this is now mentioned more explicitly.

5    **Minor Comments**

that solutions differ significantly between

1. Figure 4 caption: Is it 0.5 instead of 0.05 . It looks to me ppp aa that solutions differ significantly between $2C_p$ and $0.05C_p$

.

Answer: No, the plot is correct.

2. P6 Eq. (13) & (14): There must be a minus sign in front of the first term on the right-hand side (convergence instead of divergence).

Answer: thanks. That was a typo.

15    3. Figure 5: Does the blue curve represent the daily average temperature at 45N, S? Figure caption is confusing.

Answer: Thanks. Right. I modified the figure caption.

4. P6 L9: "The global mean temperature is not ... " It is not true. Global mean temperature from the nonlinear energy balance model used in this study will vary depending on the value of k (diffusion parameter).

20    Answer: see above at your major point 5. There is no influence of k.

5. P6 L19: "The global mean temperature ...". This is not true for the nonlinear energy balance model used in this study.

Answer: see above minor point 4.

25    6. P7 L24: less that than

Answer: I think it is ok.

**Answer to interactive comment of Referee #2 (second round) on "Temperatures from Energy Balance Models: the effective heat capacity matters"**

Gerrit Lohmann[1,2]

[1] Alfred Wegener Institute, Helmholtz Centre for Polar and Marine Research, Bremerhaven, Germany
[2] University of Bremen, Bremen, Germany

Thanks for the comment on the revised version of the manuscript *Earth Syst. Dynam. Discuss., https://doi.org/10.5194/esd-2019-35, 2019.* by referee #2.

There was a technical correction: In the third line of equation (5) on the left hand side there should not be the factor $4\pi$.

5    Answer:

Equation (5) is

$$\epsilon \sigma 4\pi \overline{T^4} = (1-\alpha)S\,\pi$$

which is correct with the definition for the average

$$\overline{T^4} = \frac{1}{4\pi} \int\limits_{-\pi/2}^{\pi/2} d\varphi \int\limits_{0}^{2\pi} \cos\varphi \quad d\Theta \quad T^4 \quad .$$

In the previous version, I have not included explicitly this definition. In the revised version, it becomes clear that (5) is correct.

[revised manuscript text omitted]

---

## Author Response (AR4)

**Answer to interactive comment of the Editor on "Temperatures from Energy Balance Models: the effective heat capacity matters"**

Gerrit Lohmann[1,2]

[1]Alfred Wegener Institute, Helmholtz Centre for Polar and Marine Research, Bremerhaven, Germany
[2] University of Bremen, Bremen, Germany

Thanks for the constructive critics on the revised version of the manuscript *Earth Syst. Dynam. Discuss., https://doi.org/10.5194/esd-2019-35, 2019.*. In the following, I repeat and answer to these comments (in blue). Furthermore, the actions are documented.

**Comment 1**

I am still not completely satisfied with the explanations behind the derivation of (10). Yes, for observational data the diurnal cycle is weak and Fig.11 shows that

$$\frac{1}{2\pi}\int_0^{2\pi} T^4 \approx \tilde{T}^4$$

but what about the solution of (9) where this assumption is used? Would you please provide the estimate for the amplitude of diurnal cycle and analogue of Fig.11 obtained for numerical solution of model (9) at some reasonable value of the heat capacity.

Answer: Thanks for this important hint. My previous statement that the diurnal cycle shall be small was not entirely correct. It is now emphasized that we assume that the exchange of operations zonal/time mean and exponentiate to the power of 4 is valid

$$\frac{1}{2\pi}\int_0^{2\pi} T^4 \approx \tilde{T}^4 \tag{1}$$

which is a good approximation in the climate system (Fig. A.1) when analyzing hourly data temperature (Hersbach et al., 2020). Indeed, the diurnal cycle amplitude could be 100 K at high latitudes, but (1) can be still valid. Fig. A.1b indicates that the difference is lowest at low latitudes or marine-dominated regions where the diurnal variation of sea surface temperatures is small (cf., Kawai and Kawamura, 2002; Stommel, 1969; Stuart-Menteth, et al. 2003; Ward, 2006).

I modified the text accordingly.

**Comment 2**

Page 1, line 1 "....simplified systems of the climate system....". Please reword.

Answer: done

**Comment 3**

Page 4, line 9. I would also add that the "local radiative equilibrium assumption" cannot be used at these conditions.

Answer: Thanks, done

**Comment 4**

3. Page 4, line 12 "elesewhere". Please correct.

Answer: Thanks, done

**Comment 5**

Page 10, line 18. I would not use "Ironically" here. Two estimates are close because the Earth is a fast rotating body with significant heat capacity having spherical geometry. Original estimate implicitly uses these facts. I do not see any irony here.

Answer: Indeed. I replaced the wording.

[revised manuscript text omitted]

---

## Author Response (AR5)

**Answer to Editor on**
**"Temperatures from Energy Balance Models: the effective heat capacity matters"**

Gerrit Lohmann[1,2]

[1]Alfred Wegener Institute, Helmholtz Centre for Polar and Marine Research, Bremerhaven, Germany
[2] University of Bremen, Bremen, Germany

Thanks for accepting the revised version of the manuscript *Earth Syst. Dynam. Discuss., https://doi.org/10.5194/esd-2019-35, 2019.*.